# A New Simplified Parameterization of Secondary Organic Aerosol in the Community Earth System Model Version 2 (CESM2; CAM6.3)

Duseong S. Jo[1], Simone Tilmes[1], and Louisa K. Emmons[1], Siyuan Wang[2,3], Francis Vitt[1]

[1]National Center for Atmospheric Research, Boulder, CO, USA
[2]Cooperative Institute for Research in Environmental Sciences, University of Colorado, Boulder, CO, USA
[3]NOAA Chemical Sciences Laboratory, Boulder, CO, USA

*Correspondence to*: Duseong S. Jo (cdswk@ucar.edu)

Manuscript submitted to Geoscientific Model Development

**Abstract.** The Community Earth System Model (CESM) community has been providing versatile modeling options, with simple to complex chemistry and aerosol schemes in a single model, in order to support the broad scientific community with various research interests. While different model configurations are available in CESM and these can be used for different fields of Earth system science, simulation results that are consistent across configurations are still desirable. Here we develop a new simple secondary organic aerosol (SOA) scheme in the Community Atmosphere Model (CAM) version 6.3, the atmospheric component of the CESM. The main purpose of this simplified SOA scheme is to reduce the differences in aerosol concentrations and radiative fluxes between CAM and CAM with detailed chemistry (CAM-chem) while maintaining the computational efficiency of CAM. CAM simulation results using the default CAM6 and the new SOA schemes are compared to CAM-chem results as a reference. More consistent SOA concentrations are obtained globally when using the new SOA scheme, for both temporal and spatial variabilities. The new SOA scheme shows 62% of grid cells globally are within a factor of 2 compared to the CAM-chem SOA concentrations, which is improved from 24% when using the default CAM6 SOA scheme. Furthermore, other carbonaceous aerosols (black carbon and primary organic aerosol) in CAM6 become closer to CAM-chem results, due to more similar microphysical aging time scales influenced by SOA coating, which in turn leads to comparable wet deposition fluxes. This results in an improved global atmospheric burden and concentrations at the high latitudes of the Northern Hemisphere compared to the full chemistry version (CAM-chem). As a consequence, the radiative flux differences between CAM-chem and CAM in the Arctic region (up to 6 W m$^{-2}$) are significantly reduced for both nudged and free-running simulations. We find that the CAM6 SOA scheme can still be used for radiative forcing calculation as the high biases exist both in pre-industrial and present conditions, but studies focusing on the instantaneous radiative effects would benefit from using the SOA scheme developed in this study. The new SOA scheme also has technical advantages including the use of identical SOA precursor emissions as CAM-chem from the online biogenic emissions, instead of pre-calculated emissions that may introduce differences. Future parameter updates on the CAM-chem SOA scheme can be easily translated to the new CAM SOA scheme as it is derived from the CAM-chem SOA scheme.

**Short Summary**. The new simple secondary organic aerosol (SOA) scheme has been developed for the Community Atmosphere Model (CAM), based on the complex SOA scheme in CAM with detailed chemistry (CAM-chem). The CAM with the new SOA scheme shows better agreements with CAM-chem in terms of aerosol concentrations and radiative fluxes, which ensures more consistent results between different compsets in the Community Earth System Model. The new SOA scheme also has technical advantages for future developments.

## 1 Introduction

Secondary organic aerosol (SOA) accounts for a substantial fraction of ambient tropospheric aerosol (Hallquist et al., 2009). Atmospheric models generally use parameterizations to simulate SOA because it is composed of a wide range of different organic molecules (Goldstein and Galbally, 2007) and due to limited knowledge of SOA formation in the atmosphere (Nault et al., 2021). The SOA parameterization in 3D atmospheric chemistry models varies from the simple method of multiplying constant yields to emissions, to the rather complex volatility basis set (VBS) approach (Donahue et al., 2006, 2011, 2012; Jimenez et al., 2009), which considers the oxidation of volatile organic compounds (VOCs) and gas-particle partitioning, as shown in the recent model intercomparison study for organic aerosol (OA) (Hodzic et al., 2020).

Climate models that have to perform hundreds of years of simulations and many ensemble members often use very simple parameterizations to calculate SOA in the model (Tsigaridis and Kanakidou, 2018), due to the high computational cost associated with chemistry, deposition, and the increased number of model tracers to be transported (Jo et al., 2019). Because SOA affects climate through aerosol-radiation and aerosol-cloud interactions, and climate also affects SOA through changing biogenic emissions and photochemistry (Gettelman et al., 2019a; Sporre et al., 2019; Tilmes et al., 2019; Jo et al., 2021), the accurate representation of SOA in climate models is important but needs to have low computational cost for long-term simulation purposes.

The Community Earth System Model Version 2 (CESM2) has two different SOA schemes, one simplified scheme for the Community Atmosphere Model (CAM) version 6 (Danabasoglu et al., 2020) and the Whole Atmosphere Community Climate Model (WACCM) version 6 with the Middle Atmosphere (MA) chemistry (Gettelman et al., 2019b), and a VBS scheme for the CAM6 with comprehensive chemistry (CAM6-chem) (Emmons et al., 2020) and the WACCM6 with the TSMLT (troposphere, stratosphere, mesosphere, and lower thermosphere) mechanism. For the purpose of climate studies using many ensemble members, CAM6 is generally used for computational efficiency. Models like WACCM6 with TSMLT are used for detailed chemistry and aerosol studies, but in general, only a few ensemble members can be performed. Ideally, the two SOA schemes in simple and complex chemistry configurations should give the same results to maintain model consistency regarding aerosol

fields and resulting climate forcings, but the spatial and temporal distributions of SOA between CAM and CAM-chem (and WACCM TSMLT) are different enough to have a significant effect on black carbon (BC) and the Earth's radiation budget (Tilmes et al., 2019).

Here we propose a new simplified and computationally affordable SOA scheme for CAM, which is based on the VBS scheme in CAM-chem. We compare three SOA schemes (VBS, simplified SOA scheme in CAM6, and the new CAM SOA scheme in this study) under a few different CESM2 configurations (specified dynamics and free-running in preindustrial and present conditions). The new approach substantially reduces the differences in aerosol and radiation values between CAM and CAM-chem (Sect. 3). The new SOA scheme also has a technical advantage as it does not need input files for the SOA precursor, but uses the same emissions files as CAM-chem or WACCM for individual SOA precursor species (isoprene, terpenes, toluene, etc.).

## 2 Method

In this section, we present SOA schemes in CAM-chem and CAM, along with the new simplified SOA scheme, as summarized in Fig. 1 and Table 1. General descriptions for other carbonaceous aerosols (BC and primary organic aerosol (POA)) are also explained as concentrations of those carbonaceous aerosols are affected by SOA concentrations (Tilmes et al., 2019). This section also includes the simulation set-up for comparisons between SOA schemes in Sect 3. To facilitate discussion throughout the paper, the existing SOA scheme used in CAM is denoted as "CAM6", and the newly developed SOA scheme in this paper is denoted as "CAM (This study)."

**Table 1.** SOA schemes used in this study. Computational costs are estimated on the Cheyenne
supercomputer at NCAR. Computational cost ranges are given in parentheses with the average value.

| SOA scheme | CAM-chem | CAM6 | CAM (This study) |
|---|---|---|---|
| Emissions | Individual VOCs, online biogenic emissions | Pre-calculated, lumped SOAG emissions | Individual VOCs, online biogenic emissions |
| VOCs and chemistry | explicitly simulated | No | Lumped tracer (SOAE) with 1-day lifetime |
| Number of SOA bins | 5 | 1 | 1 |
| Saturation vapor pressure ($\mu g\ m^{-3}$) | 0.01, 0.1, 1, 10, 100 | 1.02 | 1 |
| Enthalpy of vaporization ($kJ\ mol^{-1}$) | 153, 142, 131, 120, 109 | 156 | 131 |
| SOA yield | Based on the VBS | Fixed fraction and scaled up by 50% | Based on the VBS but lumped |
| Loss processes | wet & dry deposition of SOAG photolytic loss of soa | No deposition of SOAG No photolytic loss | wet & dry deposition of SOAG photolytic loss of soa |
| Effective Henry's law constants of SOAG ($M\ atm^{-1}$) | $4.0\times10^{11}$, $3.2\times10^{10}$, $1.6\times10^{9}$, $3.2\times10^{8}$, $1.6\times10^{7}$ | N/A | $1.6\times10^{9}$ |
| Computational cost (pe-hrs / simulated_year) | 7933 (7783 - 8083) | 2398 (2353 - 2448) | 2455 (2414 - 2501) |

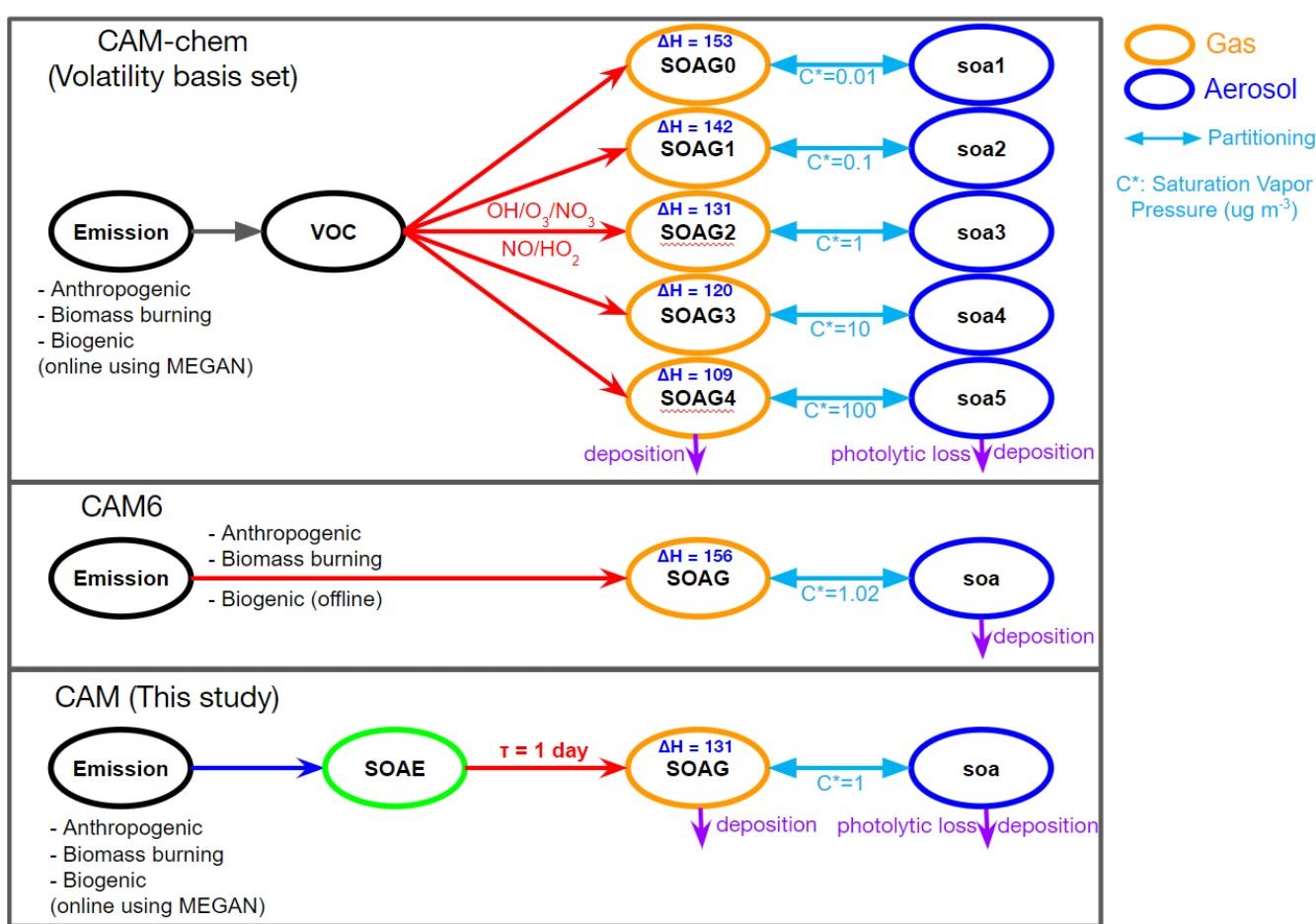

**Figure 1.** The notations are based on variable names used in CESM2. Note that "SOAG" begins with 0, while "soa" starts with 1 in CAM-chem (Tilmes et al., 2019; Emmons et al., 2020). In CESM2, gases are written in upper case and aerosols are written in lower case.

## 2.1 SOA scheme in CAM-chem

SOA in CAM-chem is simulated using the VBS approach, as described by Tilmes et al. (2019). The VBS scheme in CAM-chem incorporates recent findings such as wall-corrected SOA yields, photolytic removal of SOA, and more efficient removal by dry and wet deposition. Details can be found in Hodzic et al. (2016). The VBS approach in CAM-chem has been evaluated against surface and aircraft observations in the United States, Europe, East Asia, the Amazon, and remote atmosphere (Hodzic et al.,

2016, 2020; Tilmes et al., 2019; Jo et al., 2021; Oak et al., 2022). Here we briefly describe the characteristics that can be compared to the simple SOA scheme in CAM.

CAM-chem uses a VBS scheme with 5 volatility bins (see Fig. 1) with saturation vapor pressures spanning from 0.01 to 100 μg m$^{-3}$ at 300K. Enthalpy of vaporization values are 153, 142, 131, 120, and 109 kJ mol$^{-1}$ for 0.01, 0.1, 1, 10, and 100 μg m$^{-3}$, respectively, at 300K based on Epstein et al. (2010). Traditional SOA precursors such as isoprene, monoterpenes, sesquiterpenes, benzene, toluene, and xylenes are explicitly simulated in the model, and the oxidation of those VOCs with OH, $O_3$, and $NO_3$ makes gas phase semivolatiles (SOAG) that are in equilibrium with SOA according to the volatility bins. VOCs and oxidants are not consumed to avoid duplication, as VOC chemistry is separately simulated in CAM-chem (Jo et al., 2021). Semi- and intermediate-range volatility organic compounds (S/IVOCs) are also considered with a bimolecular OH reaction. Since S/IVOCs are defined by volatility and exact chemical speciation is not available for them, 60% of POA and 20% of total non-methane VOC (NMVOC) emissions are assumed to be SVOCs and IVOCs, respectively (Hodzic et al., 2016). Biogenic VOCs are calculated online using the Model of Emissions of Gases and Aerosols from Nature (MEGAN) version 2.1 (Guenther et al., 2012) available in the Community Land Model (CLM) version 5, a component of CESM and coupled to CAM (Lawrence et al., 2019). Photolytic removal of SOA is calculated as 0.04% of the $NO_2$ photolysis rate (Hodzic et al., 2016). Heterogeneous loss of SOA is not included in CAM-chem (Tilmes et al., 2019). However, the effect of heterogeneous removal on SOA burden is small (lifetime of 80-90 days) compared to the rapid loss of SOA due to photolysis (Hodzic et al., 2016).

CAM-chem also supports an extended VBS compset that keeps track of VBS tracers from three sources (anthropogenic, biomass burning, and biogenic), leading to 15 SOA species simulated in total. This option is not generally used except for studies tracking sources of SOA, as total SOA burden and formation are very similar between the two options because the same volatility bins are used (Tilmes et al., 2019).

In terms of aerosol modes, the four-mode version of the Modal Aerosol Module (MAM4) is generally used in recent scientific applications (Liu et al., 2016). MAM4 is a 2-moment scheme that includes interstitial and cloud-borne aerosols and considers Aitken, accumulation, coarse, and primary carbon

modes. The standard deviation of each mode is fixed, but the wet radius in each mode can change per grid box, depending on the composition. Aitken mode mass grows into the accumulation mode, and accumulation mode mass grows into the coarse mode. More details are provided in Liu et al. (2012) and (2016). SOA is simulated using Aitken and accumulation modes but most of the mass (>99%) is in the accumulation mode (Tilmes et al., 2019). In total, 15 tracers (5 for the gas phase and 10 for the aerosol phase - 5 bins × 2 modes) are used for the SOA calculation in CAM-chem.

**2.2 SOA scheme in CAM6**

The simplified SOA scheme in CAM6 uses 3 tracers (1 for the gas phase and 2 for the aerosol phase). Like the VBS, both gas-phase (SOAG) and aerosol-phase (soa_a1 and soa_a2 for accumulation and Aitken modes) are simulated with gas-aerosol partitioning, with the enthalpy of vaporization of 156 kJ mol$^{-1}$ and the saturation vapor pressure of 1.02 μg m$^{-3}$ (Liu et al., 2012). SOAG does not undergo dry and wet removal, which is also different from the VBS that calculates dry and wet deposition of gas-phase semivolatiles (SOAGs). Note that dry and wet deposition are applied to SOA in all simulation cases as shown in Fig 1.

Unlike the VBS representation which explicitly simulates parent VOCs, this scheme does not simulate the chemistry of VOCs but uses pre-calculated emissions using fixed mass yields for the following VOC categories: 5% BIGALK (lumped ≧C4 alkanes), 5% BIGENE (lumped ≧C4 alkenes), 15% aromatics, 4% isoprene, and 25% monoterpenes (Liu et al., 2012). For biogenic VOCs, offline emissions are precalculated and provided as an additional input file based on biogenic emissions simulated by CLM-MEGAN2.1. Generally, the offline biogenic VOC emission does not have annual variations and is repeated over the simulation period. Note that those SOAG emissions are further increased by 50% after model tuning involving the aerosol indirect effect (Liu et al., 2012).

**2.3 New SOA scheme in CAM**

The SOA scheme developed in this study uses a similar approach to the SOA scheme in CAM6, but several modifications have been made to allow more consistent results with the VBS scheme in CAM-chem (Fig. 1). First, VOC species that generate SOA are matched to the VBS. In other words, BIGALK

and BIGENE are no longer used for the calculation of SOA emissions, and instead, sesquiterpenes and

S/IVOCs are considered for calculating the interactive emissions of SOA. This change can be

scientifically justified because SOA yields generally increase with the carbon number (Lim and Ziemann,

2009; Srivastava et al., 2022). BIGALK and BIGENE are mainly composed of C4-C6 alkanes and alkenes

(Emmons et al., 2020), but S/IVOCs correspond to C12 or higher n-alkanes (Robinson et al., 2007).

Second, VBS product yields (forming semi-volatile compounds in the model, sum of gas and aerosol

phases, and used for the interactive emissions) have been calculated based on the CAM-chem yields,

which were adapted from Hodzic et al. (2016). The VBS product yields for the first four bins and 20% of

the fifth bin are summed up for each compound. Only 20% of the fifth bin yield is used, as it is the most

volatile bin and its saturation vapor pressure is 100 times higher than the volatility bin we use in CAM

(Fig. 1). We selected 20% based on the SOA burden comparison between CAM-chem and CAM, by

adjusting this fraction with multiple simulation tests. We consider VBS product yields from OH reactions

only in this calculation, because the reaction with OH is dominant for VOCs. Only low $NO_x$ yields are

used in this study which is consistent with Tilmes et al. (2019), which is appropriate for global climate

studies with 1° horizontal resolution of the model grid. For air quality studies with high spatial resolution,

CAM-chem with $NO_x$-dependent SOA yields can be used (Schwantes et al., 2022). The resulting yields

derived from CAM-chem results are 0.28, 0.64, 0.04, 0.16, 0.45, 0.35, 0.41, and 0.80 for monoterpenes,

sesquiterpenes, isoprene, benzene, toluene, xylenes, IVOC, and SVOC, respectively. These yields are

constants and do not change during the run, as in CAM-chem. It is worth noting that those yields can be

easily updated in the CAM run-time namelist file if there is a future update to the CAM-chem VBS

scheme.

Third, we add a new tracer called "SOAE" (Fig. 1) to consider the time that VOCs and intermediate

chemical species undergo oxidation before forming semivolatiles. We assume a constant 1-day e-folding

lifetime to convert "SOAE" to "SOAG" which can be partitioned into aerosols so that oxidant fields do

not have to be simulated in CAM for computational efficiency. The 1-day lifetime corresponds to the OH

reaction rate constant of $10^{-11}$ $cm^3$ molecules$^{-1}$ s$^{-1}$ with a global annual mean OH concentration of 11.6 ×

$10^5$ molecules $cm^{-3}$ (Warneck and Williams, 2014).

Fourth, parameters are adjusted for consistency with the VBS scheme. The enthalpy of vaporization is changed from 156 to 131 kJ mol$^{-1}$, which is the value used in the third bin of the VBS scheme. This can change SOA in the upper troposphere where temperature dependency becomes important. Deposition of gas phase semivolatiles (SOAG) and the photolytic reaction of SOA are also added (deposition of SOA is already considered in CAM6), which can affect SOA concentrations in the remote atmosphere. Saturation vapor pressure change with the assumption of 10% of POA as oxygenated (Liu et al., 2012) is not used in this scheme for consistency with the VBS scheme.

Fifth, the same offline emission files (anthropogenic and biomass burning) and online emission (biogenic) are used as the VBS method in CAM-chem, via namelist control. As a result, preprocessing for SOAG emission is no longer needed, and annual variability as well as the diurnal cycle for biogenic emission can be easily considered. Note that biogenic emission is always calculated in CLM, regardless of whether the emission is used or not in CAM or CAM-chem. Therefore, using online biogenic emissions does not add computational cost.

## 2.4 Other carbonaceous aerosols

Here we describe BC and POA simulations in CAM and CAM-chem, as those are affected by SOA concentrations through microphysics. Because BC, POA, and SOA precursors are emitted from the same sources (except for the biogenic SOA), changes in one component can significantly affect other components. Tilmes et al. (2019) reported ~20% differences between the simplified SOA and the VBS scheme in terms of the global burden of BC and POA, while the difference for the sulfate burden was very small (< 1%).

Unlike SOA, there is no difference in BC and POA simulation schemes between CAM and CAM-chem, because BC and POA are chemically inert and the standard aerosol module is the same (MAM4) for both CAM and CAM-chem. However, BC and POA can change through the following processes. Both POA and BC are emitted into the primary carbon mode, where they are coated by sulfate and SOA, and then transferred into the accumulation mode and slowly aged through condensation and coagulation, with a threshold coating thickness of eight hygroscopic monolayers of SOA (Liu et al., 2016). In the accumulation mode, aerosols are hydrophilic, with a volume-weighted hygroscopicity calculated based

on the volume mixing rule. A strong increase in SOA formation over source regions, which is true for CAM-chem SOA based on Hodzic et al. (2016) SOA scheme, increases the internally mixed aerosol number, which causes enhanced aging of BC and POA. As a result, the CAM SOA scheme simulates more than two times higher primary carbon mode concentrations of BC and POA through reduced aging, but ~10% lower accumulation mode concentrations of both. This results in increased dry deposition and decreased wet deposition in the CAM SOA scheme compared to the CAM-chem SOA scheme, as the primary carbon mode is hydrophobic but the accumulation mode is hydrophilic in CESM. More details can be found in Tilmes et al. (2019).

## 2.5 Simulation set-up

We conduct three types of model experiments for different application scenarios using the development version of CESM2.2 or CAM6.3 (tag name: cam6_3_050). First, a specified dynamics run is performed for the analysis of the present condition using the nudged meteorological fields. Temperature and horizontal winds are nudged towards the Modern-Era Retrospective analysis for Research and Applications version 2 (MERRA2) every 3 hours (Gelaro et al., 2017). In this simulation, we run the model for the year 2013 with a spin-up period of one year. Second, historical runs are performed for the 1850s and 2000s with prescribed sea surface temperatures and sea ice conditions. These are free-running simulations for 12 years for each condition with the two years discarded for the spin-up. In this case, the CLM is run with the satellite phenology (SP) option which uses a prescribed leaf area index (LAI) based on MODIS satellite observations (Lawrence et al., 2019). In this option, the input LAI value for each plant functional type (PFT) is the same between the 1850s and 2000s but the PFT fraction changes with time. As a result, the final LAI used for biogenic emission calculation is slightly different between the two periods. The third is the same as the second experiment, but the vegetation state including LAI is simulated prognostically by CLM (biogeochemistry; BGC) (Lawrence et al., 2019). In addition to absolute values, the difference between the 1850s and 2000s is investigated from the historical simulations in Sect. 3.3, to compare simulation results in terms of the radiative forcing.

In all simulations, the bi-directional oceanic flux of dimethyl sulfide (DMS) is calculated using the Online Air-Sea Interface for Soluble Species (OASISS) (Wang et al., 2019, 2020) and the climatological surface seawater DMS concentration (Lana et al., 2011), which will be the default DMS emission in the next CESM version. In brief, OASISS determines the direction and the magnitude of the ocean fluxes based on solubility, the physical conditions in the ocean (e.g., sea surface temperature, salinity, waves and bubbles) and the atmosphere (temperature, wind). Figure S1 shows the timeseries comparisons between online DMS emissions calculated by OASISS and offline DMS emissions that have been used in CAM-chem (Emmons et al., 2020). For the Northern Hemisphere winter, both emissions show similar magnitudes, but there are approximately a factor of two differences between the two emissions in other seasons. Annual mean DMS fluxes for the 1850s and 2000s are 21.6 and 22.2 TgS yr$^{-1}$ when calculated by OASISS, but are 13.8 and 13.9 TgS yr$^{-1}$ from the offline emissions. OASISS DMS emission flux is much closer to the recent global DMS emission estimates (27.1 TgS yr$^{-1}$) by Hulswar et al. (2022).

Dry deposition of aerosols is calculated using the Zhang et al. (2001) parameterization as described in Liu et al. (2012), while gas-phase compounds are dry deposited based on a resistance-based parameterization as described in Emmons et al. (2020). In CAM6, in-cloud removal in shallow convective and stratiform clouds is calculated based on the cloud and precipitation information from the MG2 microphysics scheme (Gettelman and Morrison, 2015). For wet removal in deep convective clouds, CAM6 uses the Zhang and McFarlane (1995) deep convection scheme, coupled with a unified scheme for aerosol convective transport and wet scavenging by Wang et al. (2013) with subsequent updates and improvements by Shan et al. (2021). The convective-cloud activation fractions, which are used to calculate convective in-cloud scavenging of aerosols, are set to 0.0 for the primary carbon mode and 0.8 for Aitken and accumulation modes of carbonaceous aerosols (Liu et al., 2012). Wet deposition of gaseous compounds is based on Neu and Prather (2012) with modifications by Emmons et al. (2020).

## 3 Results

In this section, the SOA scheme in CAM6 and the SOA scheme developed in this study are evaluated against CAM-chem as a reference. As SOA changes can affect the other carbonaceous aerosols and radiation fields in CESM2 (Tilmes et al., 2019), we also compare those simulation fields as shown in Table 2.

**Table 2.** Global annual mean burden of carbonaceous aerosols (SOA, SOAG (semivolatiles that are in equilibrium with particle phase SOA), BC, and POA) and radiation fields (FSNT (net shortwave flux at top of model), FLNT (net longwave flux at top of model), top of the atmosphere (TOA) imbalance, SWCF (shortwave cloud forcing), LWCF (longwave cloud forcing)). Because CAM uses the offline biogenic SOA emissions, SOA in the default CAM is not affected by the CLM option (Sect. 2.5). Units are Gg for aerosols and W m$^{-2}$ for radiation fields.

| Simulation | SOA scheme | SOA | SOAG | BC | POA | FSNT | FLNT | TOA imbalance | SWCF | LWCF |
|---|---|---|---|---|---|---|---|---|---|---|
| 2013 (Nudged) | CAM-chem | 1022 | 484 | 117 | 587 | 236.7 | 238.7 | -2.0 | -50.5 | 22.2 |
| | CAM6 | 948 | 118 | 131 | 704 | 237.7 | 239.2 | -1.5 | -49.6 | 21.7 |
| | CAM (This study) | 1027 | 129 | 111 | 574 | 237.3 | 239.3 | -2.0 | -49.8 | 21.6 |
| 1850s (SP) | CAM-chem | 780 | 367 | 31 | 299 | 232.3 | 235.0 | -2.7 | -54.6 | 26.3 |
| | CAM6 | 699 | 102 | 43 | 435 | 233.1 | 235.4 | -2.3 | -53.7 | 25.7 |
| | CAM (This study) | 747 | 94 | 30 | 300 | 232.7 | 235.3 | -2.7 | -54.0 | 25.7 |
| 2000s (SP) | CAM-chem | 793 | 375 | 89 | 510 | 231.3 | 234.0 | -2.7 | -56.1 | 25.7 |
| | CAM6 | 796 | 102 | 102 | 635 | 232.0 | 234.4 | -2.4 | -55.3 | 25.1 |
| | CAM (This study) | 744 | 105 | 83 | 488 | 231.7 | 234.4 | -2.7 | -55.6 | 25.1 |
| 1850s (BGC) | CAM-chem | 826 | 357 | 31 | 302 | 232.2 | 235.0 | -2.8 | -55.0 | 26.4 |
| | CAM (This study) | 770 | 89 | 31 | 304 | 232.6 | 235.3 | -2.7 | -54.3 | 25.9 |
| 2000s | CAM-chem | 982 | 411 | 88 | 510 | 231.3 | 234.0 | -2.7 | -56.3 | 25.8 |

| | | | | | | | | | | |
|---|---|---|---|---|---|---|---|---|---|---|
| (BGC) | CAM (This study) | 952 | 109 | 83 | 490 | 231.6 | 234.3 | -2.7 | -55.8 | 25.2 |
| 2000s - 1850s (SP) | CAM-chem | 13 | 8 | 57 | 210 | -0.98 | -0.97 | -0.01 | -1.47 | -0.54 |
| | CAM6 | 97 | 0 | 60 | 200 | -1.15 | -1.04 | -0.11 | -1.67 | -0.66 |
| | CAM (This study) | -3 | 11 | 52 | 188 | -0.98 | -0.91 | -0.07 | -1.58 | -0.70 |
| 2000s - 1850s (BGC) | CAM-chem | 156 | 54 | 57 | 208 | -0.92 | -1.08 | 0.16 | -1.31 | -0.59 |
| | CAM (This study) | 182 | 19 | 52 | 185 | -0.96 | -0.97 | 0.01 | -1.44 | -0.75 |

## 3.1 Aerosols

Table 2 shows the global annual mean burden of aerosols by different simulations, including gas-phase SOA or semivolatiles (SOAG). Two CAM cases and CAM-chem are consistent within 10% in terms of global SOA burden, with the new scheme showing better agreement. SOAG is substantially underestimated in both CAM cases, because high volatility bins (saturation vapor pressure of 10 μg m$^{-3}$ and 100 μg m$^{-3}$) are not simulated in the 1-bin simple SOA scheme. However, SOAG does not affect other aerosol concentrations and radiation fields, and therefore it is not an important species in CAM.

Although the two CAM cases show similar global SOA burdens to CAM-chem, their temporal and spatial distributions are very different. Figure 2 shows the monthly timeseries and mean vertical profile of the global SOA burden simulated by CAM and CAM-chem in 2013. In terms of reproducing CAM-chem SOA, the lower SOA during the Northern Hemisphere winter time and the SOA build-up in the upper atmosphere (< 100 hPa) are greatly improved in this study. There is still a discrepancy between CAM (this study) and CAM-chem such as SOA at around 500 hPa and at the surface (Fig. 2d), due to the limitation of using only one volatility bin in CAM (to reduce the computational cost). One fixed volatility bin with one enthalpy value cannot fully reproduce gas-phase semivolatiles simulated by five bins and the temperature dependency of volatility changes.

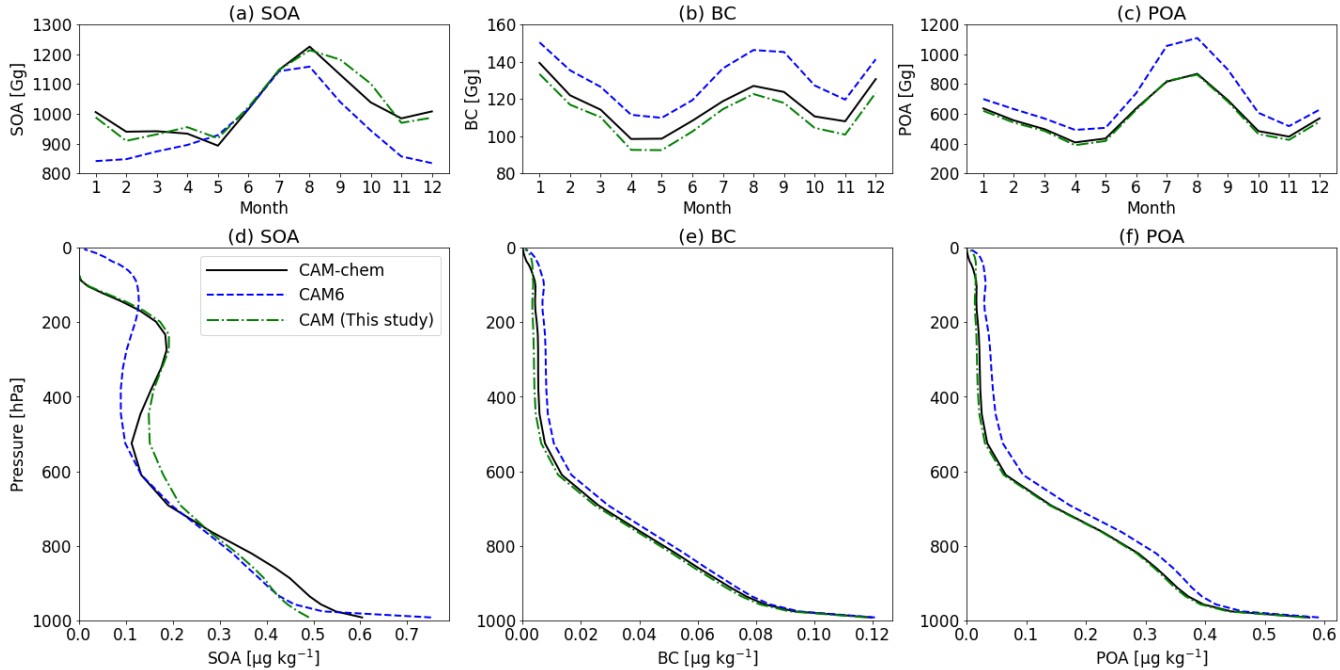

**Figure 2.** Monthly timeseries of global atmospheric burden (first row) and vertical distributions (second row) of annual average SOA, BC, and POA simulated by CESM2.

Figure 3 shows the global spatial distribution of SOA at 100 hPa, 500 hPa, 850 hPa, and the surface levels simulated by CAM-chem and CAM. In the CAM6 simulation, the main source regions (South America and Africa) are well represented at the surface layer (Fig. 3k) but do not appear in the free troposphere and above (panels b, e, and h). This is because the CAM6 SOA scheme generates semivolatiles directly from the surface emissions while the CAM-chem SOA scheme needs more time for VOC reactions to make semivolatiles, which can form SOA in the free troposphere. The intermediate tracer (SOAE) in the CAM (this study) implicitly considers this process and successfully captures SOA peaks in the free troposphere (panels c, f, and i).

In addition, the CAM6 SOA scheme fails to reproduce the sharp gradient of CAM-chem SOA above 200 hPa (Fig. 2d) and simulates too much SOA globally (Fig. 3b). The missing loss processes (deposition of semivolatiles and photolytic loss of SOA) and higher temperature dependency (enthalpy) of saturation vapor pressure result in more SOA in the CAM6 simulation. This problem is solved in the CAM SOA scheme developed in this study (Fig. 3c).

In order to quantitatively understand the relative importance of various components in the developed SOA scheme, six sensitivity simulations are conducted, as summarized in Table S1. Emission changes based on the CAM-chem VBS scheme, photolytic loss of SOA, and the intermediate tracer (SOAE) play significant roles in terms of SOA burden and similarities between CAM-chem and CAM compared to other changes made to the CAM SOA scheme described in Sect 2.3. In terms of the lifetime of SOA, both CAM-chem and CAM in this study show the same value (2.83 days) while CAM6 represents a longer lifetime (4.32 days). As a result, the fraction of grid cells within a factor of 2 and 5 compared to CAM-chem results are 62% and 82% using the CAM SOA scheme developed in this study, increased from 24% and 42% using the CAM6 scheme (Table S1). The shorter SOA lifetime in CAM-chem and CAM in this study is consistent with Hodzic et al. (2016).

Significant improvements are also found for BC and POA. CAM6 simulates up to ~45% differences while CAM in this study shows up to ~7% differences for BC and POA (Table 2). This is attributed to microphysical aging between different aerosol modes and associated wet deposition processes described in Sect 2.4. As discussed in Tilmes et al. (2019), the CAM6 SOA scheme simulates a higher primary carbon mode (41 and 276 Gg for BC and POA) compared to both CAM-chem (19 and 93 Gg) and the CAM SOA scheme in this study (14 and 81 Gg). Conversely, the CAM6 SOA scheme simulates a lower accumulation mode (90 and 429 Gg for BC and POA) compared to CAM-chem (97 and 494 Gg) and the CAM SOA scheme in this study (97 and 493 Gg).

Unlike SOA, seasonalities of BC and POA are well represented in the CAM6 (panels b and c in Fig. 2), since BC and POA schemes are the same between CAM and CAM-chem. Spatial distributions are also similar (Figs. S3–S6) except for the Arctic regions in the upper atmosphere. This difference can significantly affect the radiation budget in the Arctic region (Sect. 3.2), which should be important for climate studies focusing on the Arctic.

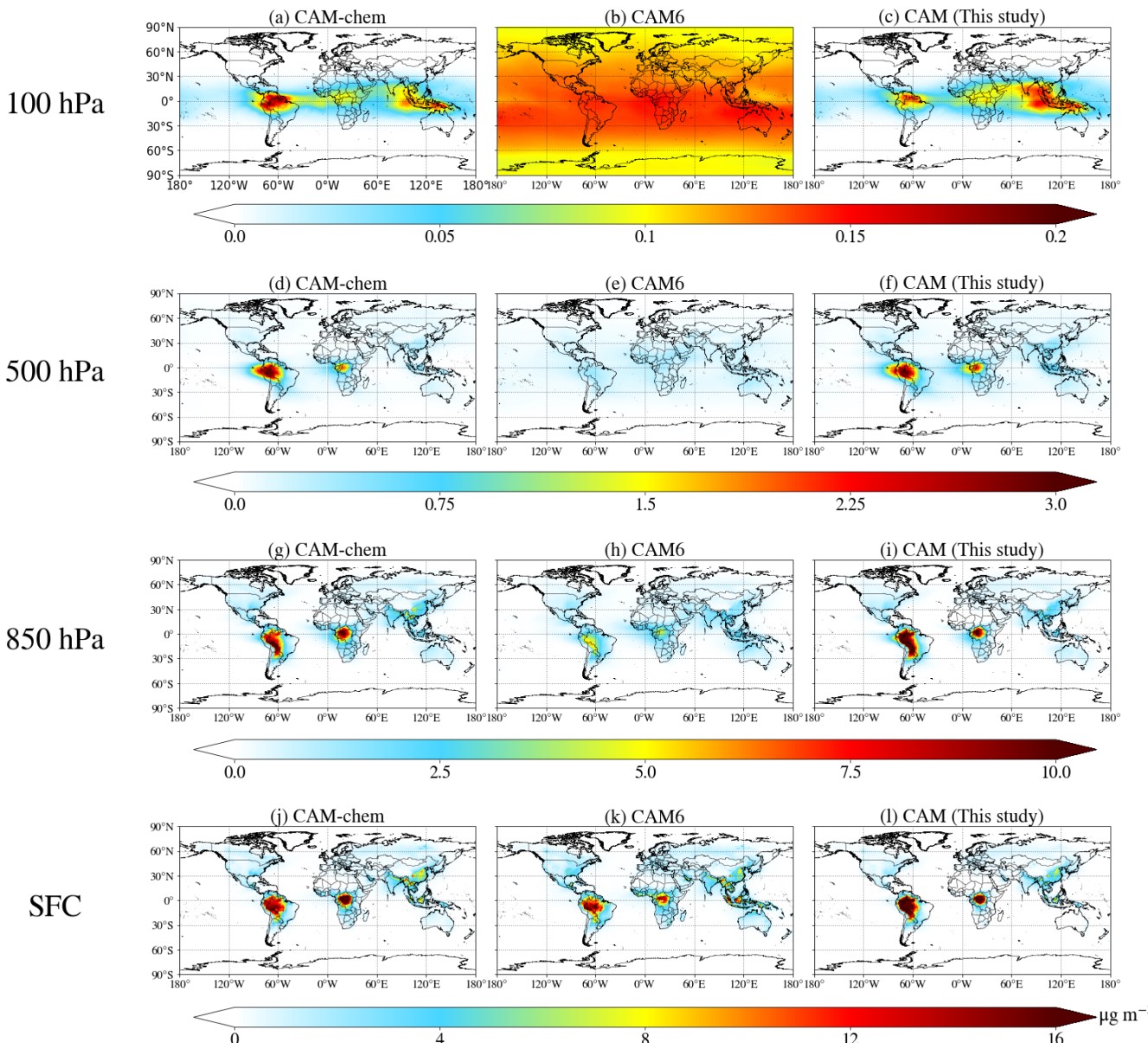

**Figure 3.** Global maps of SOA concentrations in 2013 simulated by CAM-chem (first column), CAM6 (second column), and CAM (This study) (third column) at four different vertical levels (surface, 850 hPa, 500 hPa, and 100 hPa). The difference maps between CAM and CAM-chem are available in Fig. S2.

## 3.2 Radiation fields

As aerosols can affect radiative fluxes through direct and indirect effects, here we investigate the radiation changes with the SOA scheme developed in this study, in terms of the difference between CAM and CAM-chem. Figure 4 shows the zonal averages of net shortwave (SW) and longwave (LW) fluxes and cloud forcings in CAM compared to CAM-chem. The most notable differences occur in the high latitudes in the Northern Hemisphere, similar to aerosol concentration changes shown in Sect 3.1. Both aerosol-radiation and aerosol-cloud interactions almost equally contribute to the positive bias (panels a and d). This strong positive bias of the SW flux in CAM6 is greatly improved with the SOA scheme developed in this study.

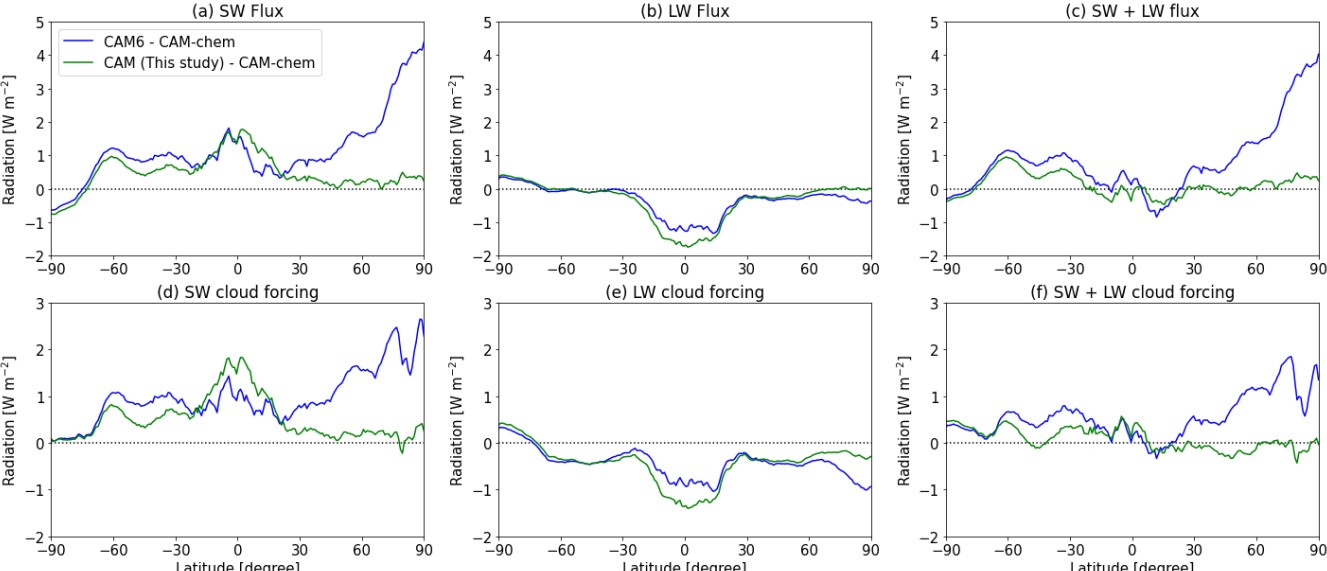

**Figure 4.** Zonal averages of the radiation difference in 2013 between CAM and CAM-chem. Radiative fluxes at the top of the model are presented in the first row (a-c) and cloud forcings are shown in the second row (d-f).

The differences between CAM-chem and CAM are slightly increased over the Tropics for individual SW and LW fluxes, which are mainly caused by cloud effects as shown in Figs. 4d and 4e, but these differences are canceled out in terms of the total radiation (Figs. 4c and 4f). Overall, the SOA scheme in this study shows slight improvements in other latitudes in addition to the Arctic region when it comes to reproducing CAM-chem results. The reduced differences can be further confirmed by the global spatial distributions shown in Fig. S7, the CAM simulation in this study shows results closer to CAM-chem in most locations globally (panels h and i in Fig. S7).

## 3.3 Historical simulations

Analogous to the simulation results with nudged meteorology in Sect 3.1 and 3.2, the SOA scheme in this study produces more consistent results with CAM-chem than the CAM6 SOA scheme (Table 2), especially for BC and POA burdens that are affected by SOA through microphysics. The new SOA scheme also captures the increased SOA burden in the 2000s compared to the 1850s when using the BGC option, which is mainly caused by increased biogenic VOC emissions (Fig. S8).

Figure S8 further shows that interannual variability may not be a significant factor for isoprene emissions on a 10-years time scale, but this would be important for climate studies with more than 100 years of simulation time (1850s vs 2000s). The offline emissions used in CAM6 have no interannual variability, thus not accounting for emission response to climate change.

The large differences between CAM-chem and CAM6 for the SW + LW flux over the Arctic from the nudged meteorology simulations (Fig. 4c) are also found in all historical simulations as shown in Fig. 5, for both 1850s and 2000s simulations. However, in terms of the difference between the 2000s and 1850s, the biases cancel out, and as a result, the difference between CAM6 and CAM-chem becomes small (Figs. 5c and 5f). This cancellation implies that previous CAM studies focusing on radiative forcing are still valid, as radiative forcing is calculated as present minus preindustrial radiative effects.

In terms of global averages (Table 2), the CAM SOA scheme in this study also demonstrates improvements in terms of consistency between CAM and CAM-chem, especially for shortwave radiation. This applies to both absolute values and the difference between present and pre-industrial simulations.

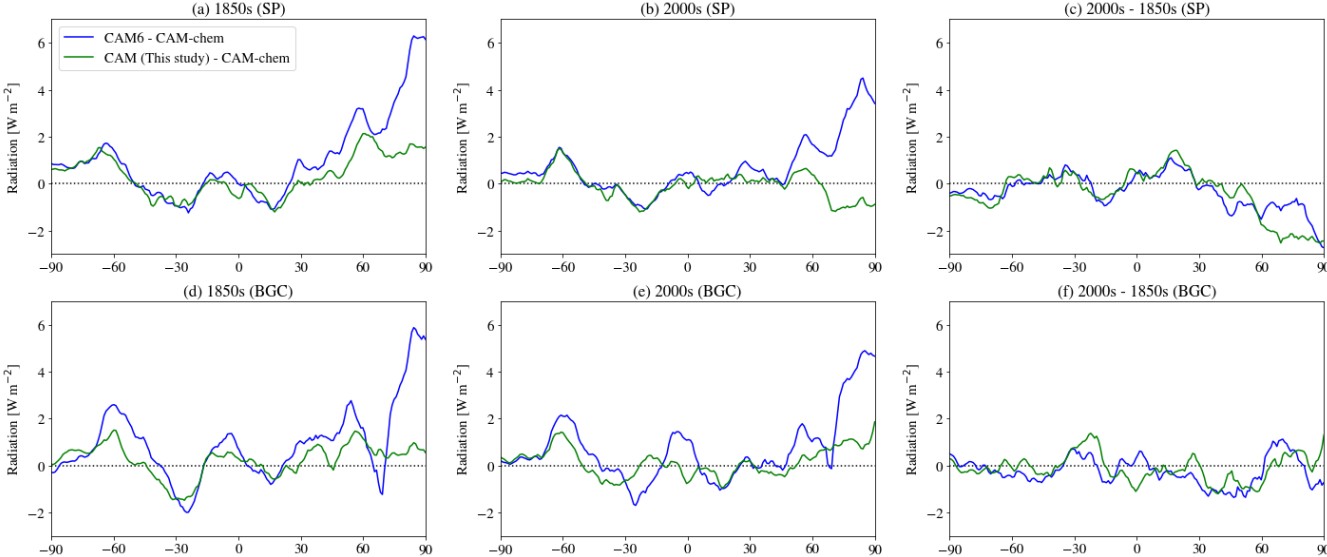

**Figure 5.** Zonal averages of the SW + LW flux difference in historical simulations (1850s (a and d), 2000s (b and e), and 2000s - 1850s (c and f)) between CAM and CAM-chem. Note that the results from the CAM6 simulations are the same for SP and BGC because CAM6 uses offline biogenic emissions. CAM-chem and CAM (This study) results affect the difference between SP and BGC simulations (blue lines).

## 4 Conclusion and possible future developments of the aerosol scheme in CAM

In this study, we developed a new SOA scheme for use in CAM with simple chemistry. This new SOA scheme was designed to close the gap between CAM and CAM-chem in terms of aerosols and radiative effects while maintaining computational efficiency. The new SOA scheme was derived based on the parameters used in the VBS scheme in CAM-chem, without changing the overall architecture of the simple SOA scheme in CAM6. For instance, VOC species for forming SOA were matched to CAM-chem, an intermediate species was introduced to mimic VOC chemistry, missing loss processes were added, and VBS parameters such as enthalpy of vaporization and saturation vapor pressure were updated.

As a result, the computational cost remained almost the same with the new SOA scheme (within the range of computing environment variability).

CAM simulation results with the two SOA schemes (CAM6 and this study) were investigated in terms of carbonaceous aerosols and radiative fluxes. There was no significant bias in terms of the global SOA burden of the CAM6 SOA scheme because it was tuned by increasing SOA emissions by 50% (Liu et al., 2012). However, the CAM6 SOA scheme was insufficient in reproducing the temporal and spatial variabilities (both horizontally and vertically) of CAM-chem SOA, while the SOA scheme in this study demonstrated similar variabilities compared to CAM-chem SOA.

The new SOA scheme also improved the simulation of other carbonaceous aerosols (BC and POA) through the microphysical processes in MAM4. Since BC and POA emissions are the same for all model cases and those aerosols are chemically inert, temporal and horizontal spatial variabilities are generally similar to each other but the absolute concentrations became closer to CAM-chem results when using the new SOA scheme. The higher BC in CAM was greatly reduced compared to CAM-chem, from ~45% in the CAM6 SOA scheme to ~7% in the new SOA scheme. POA was also improved in the same manner. Major improvements were made in the Arctic region for aerosol concentrations in the free troposphere and above.

The improvements in simulating aerosol fields led to more consistent radiative fluxes between CAM and CAM-chem, especially over the high-latitude regions in the Northern Hemisphere. The SW + LW flux at the top of the model was different by up to 6 W m$^{-2}$ and it is persistent regardless of the simulation periods in CAM6. However, in terms of radiative forcing which is calculated from the difference between present and pre-industrial conditions, both CAM6 and new CAM simulations showed no significant differences. While studies investigating instantaneous radiative effects will need to use the SOA scheme developed in this study, the CAM6 SOA scheme would still be valid for studies focusing on radiative forcing.

On the practical side, the new SOA scheme developed in this study has advantages in keeping up with the updates, as it uses the same precursor emissions as the VBS scheme in CAM-chem. The new SOA scheme uses online biogenic emissions as CAM-chem does, therefore the difference between SP

and BGC options can be calculated for SOA. If there is a future update in the VBS scheme in CAM-chem, the corresponding updates in CAM can be done easily by changing the namelist file.

Although significant advances have been made in SOA concentration simulation in this study, the aerosol module in CAM still has room for further development. Currently, CAM reads the offline monthly oxidant fields simulated by CAM-chem but oxidants such as OH and $O_3$ have strong diurnal variations. It would not be computationally feasible for CAM to calculate or read oxidants every hour, but applying constant diurnal profile values to the monthly fields would not add significant computational costs. It may be important for $SO_2$ oxidation and sulfate formation as well. The formation of SOAG from SOAE is calculated using a 1-day lifetime, but future versions could use the reaction rate constant with OH if the diurnal variation of oxidant fields is introduced in CAM. This improvement can be easily achieved by modifying the mechanism input file, however currently the prescribed OH fields are monthly means, so would provide limited improvement now.

Since there are many uncertainties in OA simulation in models, continuous updates to the CAM-chem VBS scheme will be necessary. As Hodzic et al. (2020) pointed out, CAM-chem showed good agreement in reproducing absolute OA concentrations during the Atmospheric Tomography (ATom) aircraft campaign, but the POA/SOA ratio was overestimated. CAM-chem considers SOA from S/IVOCs based on the assumption that the emission inventory they used reported POA emissions after evaporation to S/IVOCs (Hodzic et al., 2016). However, there is a possibility of double-counting depending on the timing of measuring POA emission flux. Additionally, the assumption that SVOC emissions were included in POA emissions was not sufficiently constrained due to limited observation data (Wu et al., 2019). Fang et al. (2021) reported that IVOCs did not show significant correlations with POA or NMVOCs for on-road vehicles. CAM-chem also assumes a single value for the organic mass to organic carbon (OM/OC) ratio of 1.4 for POA. In contrast, GEOS-Chem has used an OM/OC ratio of 2.1 for POA (Henze et al., 2008; Jo et al., 2013; Hodzic et al., 2020), which would lead to 50% higher POA concentrations than CAM-chem if other conditions are the same. However, observed OM/OC values are spatially and seasonally dependent, typically ranging from 1.3 to 2.5 (Aiken et al., 2008; Philip et al., 2014). These uncertain factors suggest that current assumptions about S/IVOCs and POA may need to be

updated in the future. Still, such updates in CAM-chem can be easily transferred into CAM through the consistent framework established in this study.

The SOA scheme in this study can be further adjusted depending on the research interest. For example, for studies focusing on surface aerosol fields, users can easily modify SOA yields for different emission sources through namelist changes. For studies focusing on urban air quality and resulting climate effects, SOA yields can be changed to high-$NO_x$ yields instead of low-$NO_x$ yields without code changes. Vertical shapes can be also adjusted by changing the parameters such as the enthalpy of vaporization, saturation vapor pressure, and photolysis rates in the future.

**Code and data availability.** CESM is an open-source community model and is publicly available at: https://github.com/ESCOMP/CESM. The new SOA scheme is included in the development version of CAM (https://github.com/ESCOMP/CAM, tag name: cam6_3_093) and is also available at the Zenodo repository (https://doi.org/10.5281/zenodo.7807711), and will be publicly available in the next CESM release (CESM3). The model results used in this study are available on Zenodo (https://doi.org/10.5281/zenodo.8044704).

**Author contributions.** DSJ, ST, and LKE designed the research and developed the SOA scheme. SW developed the OASISS scheme. DSJ, ST, and FV conducted CESM simulations. DSJ wrote the manuscript. All authors contributed to editing the manuscript.

**Competing interests.** The authors declare that they have no conflict of interest.

**Acknowledgments**. This material is based upon work supported by the National Center for Atmospheric Research, which is a major facility sponsored by the National Science Foundation (NSF) under Cooperative Agreement No. 1852977. This research was supported by NASA ACCDAM (award 80NSSC21K1439). We would like to acknowledge high-performance computing support from Cheyenne (doi:10.5065/D6RX99HX) provided by NCAR's Computational and Information Systems Laboratory,

sponsored by the NSF. The authors thank Behrooz Roozitalab and Alma Hodzic (NCAR) for their

valuable comments on the manuscript.

500

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
