# Peer review of "A New Simplified Parameterization of Secondary Organic Aerosol in"

_Geoscientific Model Development, 2023_

## Author Comment (AC2)

We would like to thank the reviewers for their time and valuable comments that have helped to improve our paper. To guide the review process, comments from reviewers are in black, responses in blue, and new texts added to the paper in dark green. Links are provided below for easy navigation in the document.

**General**

Reviewer #1

Reviewer #2

Reviewer #3

Reviewer #4

**General**

We have added a supplementary Figure S1 to compare DMS emissions from OASISS with offline emissions that have been used in CAM-chem and CAM. Although this is not relevant to the main points of this study, it was deemed valuable to document it for future CESM users, so Figure S1 has been added.

The following text and a figure have been added to the method section.

In brief, this model determines the direction and the magnitude of the ocean fluxes based on solubility, the physical conditions in the ocean (e.g., sea surface temperature, salinity, waves and bubbles) and the atmosphere (temperature, wind). Figure S1 shows the timeseries comparisons between online DMS emissions calculated by OASISS and offline DMS emissions that have been used in CAM-chem (Emmons et al., 2020). For the Northern Hemisphere winter, both emissions show similar magnitudes, but there are approximately a factor of two differences between the two emissions in other seasons. Annual mean DMS fluxes for the 1850s and 2000s are 21.6 and 22.2 TgS yr-1 when calculated by OASISS, but are 13.8 and 13.9 TgS yr-1 from the offline emissions. OASISS DMS emission flux is much closer to the recent global DMS emission estimates (27.1 TgS yr-1) by Hulswar et al. (2022).

Figure S1. Simulated (OASISS) and offline DMS emission timeseries in the 1850s and 2000s.

**Reviewer #1**

1.1) This manuscript provides in a compact way a new module to describe secondary organic aerosol concentrations, and its impact on radiation in a simple, and computationally efficient way, using the complex parameterization as in CAM-chem as a reference. This replaces a previous simplistic implementation which has known limitations, with only a marginal increase in computational costs. Furthermore, it allows a more closer alignment with the full-chemistry parameterization in future.

The authors present in a clear way the improvement that is gained compared to the original module with respect to the reference (full chemistry) configuration from a climatological perspective, i.e. the main application area of this new module.

While reading the manuscript, I wondered why a single 1-day e-folding lifetime for the SOAE is used, (line 166) as I believe it would make much more sense (without a significant increase in computational cost) to make this loss rate dependant on the seasonal (and diurnal) change in OH. This could lead to some changes in the seasonal cycle of the SOA burden, and its profile, I can imagine. But the authors have also noticed this as a point of future improvement.

This manuscript is fit for publication in GMD, if possible after consideration of the following comment. Table 2: Can the authors report on the changes in SOA lifetime (or SOA production) across the different configurations? If this does not fit in the table, this could also be reported as part of the text for the 2013-experiments only.

As the reviewer mentioned, making SOAE loss dependent on OH concentration is an area for future improvement, but beyond the scope of this work. Although it is not complex to include this, CAM currently does not include diurnal changes of OH, which will require greater effort. Once CAM has the capability to incorporate diurnal OH variations in the model, it will be possible to switch from a 1-day e-folding lifetime to an OH-dependent reaction.

We have added the text to clarify this as follows.

This improvement can be easily achieved by modifying the mechanism input file, however currently the prescribed OH fields are monthly means, so would provide limited improvement now.

We have calculated SOA loss processes and lifetime for 2013 experiments. As the reviewer mentioned, those values do not fit in Table 2, therefore we have made a new Table S1 as follows. Note that we have also added statistics and other sensitivity results in Table S1, based on the suggestions by reviewer #2.

**Table S1.** Global annual budgets and statistics for simulated SOA. Results are based on one year nudged simulation in 2013. The table also includes sensitivity simulation results that demonstrate the effect of excluding a specific aspect of the SOA scheme developed in this study. Three statistics are calculated against CAM-chem results: Normalized Mean Bias (NMB), the fraction of grid cells within a factor of 2 (FO2) and 5 (FO5). Statistics are calculated based on monthly mean grid cell points.

| Simulation case                                                     | Burden
(Gg) | All Loss
(Tg yr -1 ) | Dry dep.
(Tg yr -1 ) | Wet dep.
(Tg yr -1 ) | Photo.
loss
(Tg yr -1 ) | Lifetime
(days) | NMB
(%) | FO2
(%) | FO5
(%) |
|---------------------------------------------------------------------|----------------|------------------------------------|------------------------------------|------------------------------------|------------------------------------------|--------------------|------------|------------|------------|
| CAM-chem                                                            | 1022           | 132                                | 10.1                               | 66.0                               | 55.9                                     | 2.83               | -          | -          | -          |
| CAM6                                                                | 948            | 80                                 | 11.9                               | 68.2                               | 0.0                                      | 4.32               | -7.2       | 24         | 44         |
| CAM
(this study)                                                 | 1027           | 133                                | 7.8                                | 67.5                               | 57.3                                     | 2.83               | 0.5        | 62         | 82         |
| This study
(with CAM6 SOAG
emissions)                         | 318            | 37                                 | 2.4                                | 15.7                               | 18.8                                     | 3.14               | -68.9      | 32         | 70         |
| This study
(without photolytic loss)                             | 2997           | 116                                | 10.5                               | 105.3                              | 0.0                                      | 9.45               | 193.2      | 14         | 27         |
| This study
(with CAM6 saturation vapor
pressure and enthalpy) | 1057           | 138                                | 7.9                                | 70.8                               | 58.8                                     | 2.81               | 3.4        | 60         | 82         |
| This study
(without deposition of SOAG)                          | 1367           | 194                                | 11.3                               | 109.0                              | 74.2                                     | 2.57               | 33.7       | 61         | 86         |
| This study
(with the assumption of 10%
of POA as oxygenated)  | 1126           | 149                                | 9.2                                | 77.6                               | 61.9                                     | 2.77               | 10.2       | 63         | 83         |
| This study
(without intermediate tracer
SOAE)                 | 714            | 117                                | 13.6                               | 69.8                               | 33.9                                     | 2.22               | -30.1      | 16         | 39         |

We have also added the text in the manuscript as follows:

In terms of the lifetime of SOA, both CAM-chem and CAM in this study show the same value (2.83 days) while CAM6 represents a longer lifetime (4.32 days).

The shorter SOA lifetime in CAM-chem and CAM in this study is consistent with Hodzic et al. (2016).

Technical comments:

1.2) Figure 1, CAM-chem description: Here SOAG0 is in balance with SOA1, shouldn't this be 'SOA0' (and likewise SOAG1 and SOAG2)?

We understand that this notation can be confusing, but we would like to follow the variable names that have been used in CAM-chem (Tilmes et al., 2019; Emmons et al., 2020). Because changing numbers could cause another confusion for users familiar with CAM-chem, we have decided to retain this notation in this paper. We have clarified the notation for readers in Figure 1 caption, as follows.

Figure 1. Schematic diagrams of SOA parameterizations in CESM2. The notations are based on variable names used in CESM2. Note that "SOAG" begins with 0, while "soa" starts with 1 in CAM-chem (Tilmes et al., 2019; Emmons et al., 2020). In CESM2, gases are written in upper case and aerosols are written in lower case.

1.3) line 143: "after model tuning involving the aerosol indirect effect"

We have corrected this phrase.

1.4) line 260: "although the two CAM cases"

We have corrected this phrase.

**Reviewer #2**

Jo and coauthors present an improvement to the CAM6 model that updates the treatment of SOA so that it is more consistent with the predictions of CAM-Chem. The team has chosen their updates judiciously and demonstrated impressive consistency in results for pollutant concentrations and radiative forcing compared with the CAM-Chem simulations. Important divergences like SOA at the top of the atmosphere seem to be almost completely resolved. The manuscript is organized well, and the figures capably illustrate the main points of the discussion. This paper should be published in GMD. I have a number of concerns I would like the authors to address first though.

2.1) The main criticism is that the discussion of results speculates often about the aspect of the update that is responsible for a particular improvement in results. For example, lines 271-278 attribute the improvement in vertical profile to the incorporation of the intermediate tracer and introduction of a time delay in SOA formation. But part of this could also be due to the change in deposition or enthalpy of vaporization. Because there are a limited number of parameters that describe the new SOA configuration, it would be instructive to implement them one at a time and present the influence each has incrementally on the final results.

We agree that it is important to quantitatively calculate and show which of the various treatments introduced in this study had the greatest impact on the changes in SOA concentration in CAM. Therefore, we have conducted six sensitivity simulations by removing a specific aspect(s) of SOA scheme developed in this study:

- a) without changing emissions (i.e. using the current precalculated CAM6 emissions for lumped semivolatiles)
- b) without photolytic loss of SOA
- c) without changing saturation vapor pressure and enthalpy of vaporization
- d) without wet and dry deposition of semivolatile gases (i.e. SOAG in CAM)
- e) without removing the assumption of 10% POA as oxygenated
- f) without intermediate tracer SOAE

Results are summarized with budgets and similarity statistics against CAM-chem results in Table S1. We have found that three aspects (a,b,f) significantly changed SOA burden in terms of agreement (within a factor of 2 and 5) between CAM-chem and CAM, while the other three (c,d,e) were relatively less important.

**Table S1.** Global annual budgets and statistics for simulated SOA. Results are based on one year nudged simulation in 2013. The table also includes sensitivity simulation results that demonstrate the effect of excluding a specific aspect of the SOA scheme developed in this study. Three statistics are calculated against CAM-chem results: Normalized Mean Bias (NMB), the fraction of grid cells within a factor of 2 (FO2) and 5 (FO5). Statistics are calculated based on monthly mean grid cell points.

| Simulation case                                                     | Burden
(Gg) | All Loss
(Tg yr -1 ) | Dry dep.
(Tg yr -1 ) | Wet dep.
(Tg yr -1 ) | Photo.
loss
(Tg yr -1 ) | Lifetime
(days) | NMB
(%) | FO2
(%) | FO5
(%) |
|---------------------------------------------------------------------|----------------|------------------------------------|------------------------------------|------------------------------------|------------------------------------------|--------------------|------------|------------|------------|
| CAM-chem                                                            | 1022           | 132                                | 10.1                               | 66.0                               | 55.9                                     | 2.83               | -          | -          | -          |
| CAM6                                                                | 948            | 80                                 | 11.9                               | 68.2                               | 0.0                                      | 4.32               | -7.2       | 24         | 44         |
| CAM
(this study)                                                 | 1027           | 133                                | 7.8                                | 67.5                               | 57.3                                     | 2.83               | 0.5        | 62         | 82         |
| This study
(with CAM6 SOAG
emissions)                         | 318            | 37                                 | 2.4                                | 15.7                               | 18.8                                     | 3.14               | -68.9      | 32         | 70         |
| This study
(without photolytic loss)                             | 2997           | 116                                | 10.5                               | 105.3                              | 0.0                                      | 9.45               | 193.2      | 14         | 27         |
| This study
(with CAM6 saturation vapor
pressure and enthalpy) | 1057           | 138                                | 7.9                                | 70.8                               | 58.8                                     | 2.81               | 3.4        | 60         | 82         |
| This study
(without deposition of SOAG)                          | 1367           | 194                                | 11.3                               | 109.0                              | 74.2                                     | 2.57               | 33.7       | 61         | 86         |
| This study
(with the assumption of 10%
of POA as oxygenated)  | 1126           | 149                                | 9.2                                | 77.6                               | 61.9                                     | 2.77               | 10.2       | 63         | 83         |
| This study
(without intermediate tracer
SOAE)                 | 714            | 117                                | 13.6                               | 69.8                               | 33.9                                     | 2.22               | -30.1      | 16         | 39         |

We have also added the related discussion as follows:

In order to quantitatively understand the relative importance of various components in the developed SOA scheme, six sensitivity simulations are conducted, as summarized in Table S1. Emission changes based on the CAM-chem VBS scheme, photolytic loss of SOA, and the intermediate tracer (SOAE) play significant roles in terms of SOA burden and similarities between CAM-chem and CAM compared to other changes made to the CAM SOA scheme described in Sect 2.3. In terms of the lifetime of SOA, both CAM-chem and CAM in this study show the same value (2.83 days) while CAM6 represents a longer lifetime (4.32 days). As a result, the fraction of grid cells within a factor of 2 and 5 compared to CAM-chem results are 62% and 82% using the CAM SOA scheme developed in this study, increased from 24% and 42% using the CAM6 scheme (Table S1). The shorter SOA lifetime in CAM-chem and CAM in this study is consistent with Hodzic et al. (2016).

2.2) The abstract lacks any quantitative information documenting the improvement in predictive power of CAM with the new SOA scheme. Please add some summarizing statistics.

We have calculated the similarity statistics in Table S1 (the response above) and added the text in the abstract as follows:

The new SOA scheme shows 62% of grid cells globally are within a factor of 2 compared to the CAM-chem SOA concentrations, which is improved from 24% when using the default CAM6 SOA scheme.

2.3) Can the authors please add more information about the deposition scheme and parameters used for the loss of aerosol- and, particularly, gas-phase SOA species? Are Henry's Law values parameterized and used? Is pH considered?

We have added more information in Table 1 (Henry's law values) and the main text (loss processes for gases and aerosols) as follows.

[revised manuscript text omitted]

2.4) What is the sensitivity of results to changes in the assumed saturation concentration and enthalpy of vaporization?

This is one of the sensitivity simulations we have conducted, as detailed in response 2.1. To summarize, the changes in SOA due to saturation vapor pressure and enthalpy of vaporization were small compared to the changes by other parameter adjustments. More details can be found in response 2.1.

2.5) Section 2.4: Why are BC and POA particles emitted in the accumulation mode instead of the Aitken mode where most particles are emitted by anthropogenic combustion sources? What is the sensitivity to this assumption?

BC and POA particles are emitted in the primary carbon mode, which is a separate mode exclusively for these two aerosol types. As shown in Table 1 from Liu et al. (2012), typical 10th and 90th percentile size ranges in CESM for Aitken, accumulation, and primary carbon modes are approximately 0.015-0.052  $\mu$ m, 0.056-0.26  $\mu$ m, and 0.039-0.13  $\mu$ m, respectively. The primary carbon mode lies between the Aitken and accumulation mode ranges and moves to the accumulation mode through condensation and coagulation processes. Therefore, Aitken mode is not needed for the simulation of BC and POA in CESM.

2.6) Section 2.4: If SVOCs and IVOCs are added to the model, can the authors explain how they know they aren't double-counting carbon emissions with POA?

S/IVOCs emissions and SOA formation from these are the subjects of the SOA scheme in CAM-chem. There are uncertainties and limitations with CAM-chem OA simulation methods, but we would like to mention that these are beyond the scope of this study, which aims to develop a new SOA framework in CAM to achieve consistent results between CAM and CAM-chem. Additionally, future improvements in CAM-chem can be easily incorporated into CAM using the method in this study.

However, we understand the reviewer's concern about this double-counting issue in CAM-chem, as it will affect CAM simulations too, since we try to make CAM SOA similar to CAM-chem SOA. Therefore we have decided to add a detailed discussion about S/IVOCs and POA to the manuscript, as follows.

Since there are many uncertainties in OA simulation in models, continuous updates to the CAM-chem VBS scheme will be necessary. As Hodzic et al. (2020) pointed out, CAM-chem showed good agreement in reproducing absolute OA concentrations during the Atmospheric

Tomography (ATom) aircraft campaign, but the POA/SOA ratio was overestimated. CAM-chem considers SOA from S/IVOCs based on the assumption that the emission inventory they used reported POA emissions after evaporation to S/IVOCs (Hodzic et al., 2016). However, there is a possibility of double-counting depending on the timing of measuring POA emission flux. Additionally, the assumption that SVOC emissions were included in POA emissions was not sufficiently constrained due to limited observation data (Wu et al., 2019). Fang et al. (2021) reported that IVOCs did not show significant correlations with POA or NMVOCs for on-road vehicles. CAM-chem also assumes a single value for the organic mass to organic carbon (OM/OC) ratio of 1.4 for POA. In contrast, GEOS-Chem has used an OM/OC ratio of 2.1 for POA (Henze et al., 2008; Jo et al., 2013; Hodzic et al., 2020), which would lead to 50% higher POA concentrations than CAM-chem if other conditions are the same. However, observed OM/OC values are spatially and seasonally dependent, typically ranging from 1.3 to 2.5 (Aiken et al., 2008; Philip et al., 2014). These uncertain factors suggest that current assumptions about S/IVOCs and POA may need to be updated in the future. Still, such updates in CAM-chem can be easily transferred into CAM through the consistent framework established in this study.

2.7) Section 2.4: It is surprising to me that the differences in Figs. 2b and 2c are due entirely to SOA aging. Can the authors be a little more specific about the mechanism of the change in deposition for POA and BC? Is this an impact of particle size change or change in bulk particle hydrophilicity?

It was originally given in detail by Tilmes et al. (2019), as this study first showed the differences in POM and BC burdens resulting from the choice of either the CAM-chem or CAM6 SOA scheme. We have now added a comprehensive discussion on how POA and BC are calculated with two different aerosol modes, including aging and deposition with different hygroscopicities, in Section 2.4 as follows.

Unlike SOA, there is no difference in BC and POA simulation schemes between CAM and CAM-chem, because BC and POA are chemically inert and the standard aerosol module is the same (MAM4) for both CAM and CAM-chem. However, BC and POA can change through the following processes. Both POA and BC are emitted into the primary carbon mode, where they are coated by sulfate and SOA, and then transferred into the accumulation mode and slowly aged through condensation and coagulation, with a threshold coating thickness of eight hygroscopic monolayers of SOA (Liu et al., 2016). In the accumulation mode, aerosols are hydrophilic, with a volume-weighted hygroscopicity calculated based on the volume mixing rule. A strong increase in SOA formation over source regions, which is true for CAM-chem SOA based on Hodzic et al. (2016) SOA scheme, increases the internally mixed aerosol number, which causes enhanced aging of BC and POA. As a result, the CAM SOA scheme simulates more than two times higher primary carbon mode concentrations of BC and POA through reduced aging, but ~10% lower

accumulation mode concentrations of both. This results in increased dry deposition and decreased wet deposition in the CAM SOA scheme compared to the CAM-chem SOA scheme, as the primary carbon mode is hydrophobic but the accumulation mode is hydrophilic in CESM. More details can be found in Tilmes et al. (2019).

We have also added the related discussion in Sect. 3.1.

Significant improvements are also found for BC and POA. CAM6 simulates up to ~45% differences while CAM in this study shows up to ~7% differences for BC and POA (Table 2). This is attributed to microphysical aging between different aerosol modes and associated wet deposition processes described in Sect 2.4. As discussed in Tilmes et al. (2019), the CAM6 SOA scheme simulates a higher primary carbon mode (41 and 276 Gg for BC and POA) compared to both CAM-chem (19 and 93 Gg) and the CAM SOA scheme in this study (14 and 81 Gg). Conversely, the CAM6 SOA scheme simulates a lower accumulation mode (90 and 429 Gg for BC and POA) compared to CAM-chem (97 and 494 Gg) and the CAM SOA scheme in this study (97 and 493 Gg).

2.8) Please add details to the methods description explaining what MAM4 is using for aerosol moments (e.g. 2-moment or 3-moment). Is the standard deviation of each mode a variable parameter? Also, does Aitken mode mass grow into the Accumulation mode?

We have clarified these details in the method section as follows.

MAM4 is a 2-moment scheme that includes interstitial and cloud-borne aerosols and considers Aitken, accumulation, coarse, and primary carbon modes. The standard deviation of each mode is fixed, but the wet radius in each mode can change per grid box, depending on the composition. Aitken mode mass grows into the accumulation mode, and accumulation mode mass grows into the coarse mode. More details are provided in Liu et al. (2012) and (2016).

2.9) Lines 154-162: More details should be given about how the yields were parameterized for the new SOA scheme. It is clear that the yields for the c\* = 100 ug m-3 bin should be reduced, but how did the authors choose 20% and what is the sensitivity of results to this parameter? Early in this paragraph, it is claimed that the yields for the four lowest bins are merely added up, but later the authors write that the yields are based on CAM-Chem results. Which of these are true, and what is the basis for the choice?

Our SOA yields are based on CAM-chem VBS yields, but the last bin was reduced as our SOA scheme only uses one volatility bin. Adding up SOA yields across all volatility bins in CAM-chem will lead to overestimation, as SOA in the last volatility bin ( $C^* = 100$ )

 $\mu$ g m-3) should be mainly in the gas phase but the CAM SOA bin partitions more aerosols with C\* = 1  $\mu$ g m-3. We chose a 20% for the last (fifth) bin based on the global burden comparison between CAM-chem and CAM, by conducting multiple sensitivity simulations by changing the fraction of the last bin. We have clarified this in the text as follows.

Only 20% of the fifth bin yield is used, as it is the most volatile bin and its saturation vapor pressure is 100 times higher than the volatility bin we use in CAM (Fig. 1). We selected 20% based on the SOA burden comparison between CAM-chem and CAM, by adjusting this fraction with multiple simulation tests.

2.10) How sensitive are these results to the choice of aerosol activation scheme and cloud microphysics module?

We have not tested the sensitivity to the aerosol activation scheme or cloud microphysical module. However, we do not expect any dependencies, since our approach has been developed based on CAM-chem, which uses the same scheme as in CAM. In addition, we use the microphysics module (MG2; Gettelman and Morrison, 2015) in this study, which has been used for most CESM application studies.

Gettelman, A. and Morrison, H.: Advanced Two-Moment Bulk Microphysics for Global Models. Part I: Off-Line Tests and Comparison with Other Schemes, J. Clim., 28(3), 1268–1287, 2015.

Minor Comments/Typos:

2.11) Lines 43-45: It seems odd to introduce the VBS without citing any of the formative VBS literature references (e.g. Donahue et al., 2006, 2011, 2012; Jimenez et al., 2009, etc.)

We have added the references suggested as follows.

The SOA parameterization in atmospheric chemistry models varies from the simple method of multiplying constant yields to emissions, to the complex volatility basis set (VBS) approach (Donahue et al., 2006, 2011, 2012; Jimenez et al., 2009),

2.12) Line 63: 'the same' --> 'similar'

We believe it would be better to use "same" instead of "similar" as we are discussing an ideal case here.

2.13) Line 112: 'simple' --> 'first-order' or 'bimolecular'

We have changed it to bimolecular.

2.14) Line 150: Although it is cited in Shrivastava et al. (2022), I recommend adding a specific reference to Lim and Ziemann (2009) since they showed this definitively.

We have added Lim and Ziemann (2009).

**Reviewer #3**

This paper presents a new simplified secondary organic aerosol scheme in the atmospheric component of CESM2 model (CAM6.3). The new approach aims to replace the previous simplified SOA formation scheme available in CAM with known biases compared with the more complex parameterization based on the volatility basis set (VBS) approach (CAM-chem). The authors describe the new scheme and the design strategies adopted to achieve closer results to CAM-Chem at a much lower computational cost. The paper is well structured and the figures and tables provide proper support to the description of the work. However, some additional details could be provided for a comprehensive description of the new scheme presented in the manuscript. This paper should be accepted for publication in Geoscientific Model Development after addressing the following minor comments.

**Specific comments:**

3.1) The authors could provide some quantitative results in the abstract instead of using expressions like "the overestimation ... is greatly reduced" or " the high bias ... is significantly reduced".

We have added the following text in the abstract as follows.

The new SOA scheme shows 62% of grid cells globally are within a factor of 2 compared to the CAM-chem SOA concentrations, which is improved from 24% when using the default CAM6 SOA scheme.

As a consequence, the radiative flux differences between CAM-chem and CAM in the Arctic region (up to  $6 \text{ W m}^{-2}$ ) are significantly reduced for both nudged and free-running simulations.

3.2) Provide some key parameters used in the new scheme for the sake of completeness in Table 1: (1) the Henry's law constants of SOA gas-phase precursors used in the new scheme (is the ones described in Hodzic et al. (2016) Table 2 for the third volatility bin?), (2) the wet scavenging efficiency of SOA (set to 80%?), (3) the photolysis rate coefficient (set to 0.04% of NO2 one?), and (4) the uptake coefficient of the heterogeneous reaction with ozone (set to 10e-5).

We have updated Table 1 to include the effective Henry's law constants for CAM-chem and CAM. For other variables, those are universally applied to all model cases, therefore we have included the descriptions in the text. Photolytic removal of SOA is calculated as 0.04% of the NO2 photolysis rate (Hodzic et al., 2016). Heterogeneous loss of SOA is not included in CAM-chem (Tilmes et al., 2019). However, the effect of heterogeneous removal on SOA burden is small (lifetime of 80-90 days) compared to the rapid loss of SOA due to photolysis (Hodzic et al., 2016).

The convective-cloud activation fractions, which are used to calculate convective in-cloud scavenging of aerosols, are set to 0.0 for the primary carbon mode and 0.8 for Aitken and accumulation modes of carbonaceous aerosols (Liu et al., 2012).

**Table 1.** SOA schemes used in this study. Computational costs are estimated on the Cheyenne supercomputer at NCAR. Computational cost ranges are given in parentheses with the average value.

| SOA scheme                                                           | CAM-chem                                                                                                            | CAM6                                        | CAM (This study)                                          |
|----------------------------------------------------------------------|---------------------------------------------------------------------------------------------------------------------|---------------------------------------------|-----------------------------------------------------------|
| Emissions                                                            | Individual VOCs, online
biogenic emissions                                                                       | Pre-calculated, lumped
SOAG emissions    | Individual VOCs, online biogenic emissions                |
| VOCs and chemistry                                                   | explicitly simulated                                                                                                | No                                          | Lumped tracer (SOAE)
with 1-day lifetime               |
| Number of SOA bins                                                   | 5                                                                                                                   | 1                                           | 1                                                         |
| Saturation vapor pressure
(µg m -3 )                   | 0.01, 0.1, 1, 10, 100                                                                                               | 1.02                                        | 1                                                         |
| Enthalpy of vaporization
(kJ mol -1 )                  | 153, 142, 131, 120, 109                                                                                             | 156                                         | 131                                                       |
| SOA yield                                                            | Based on the VBS                                                                                                    | Fixed fraction and scaled up by 50%         | Based on the VBS
but lumped                            |
| Loss processes                                                       | wet & dry deposition of
SOAG
photolytic loss of soa                                                           | No deposition of SOAG
No photolytic loss | wet & dry deposition of
SOAG
photolytic loss of soa |
| Effective Henry's law
constants of SOAG
(M atm -1 ) | 4.0×10 11 , 3.2×10 10 ,
1.6×10 9 , 3.2×10 8 ,
1.6×10 7 | N/A                                         | 1.6×10 9                                       |
| Computational cost
(pe-hrs / simulated_year)                      | 7933 (7783 - 8083)                                                                                                  | 2398 (2353 - 2448)                          | 2455 (2414 - 2501)                                        |

3.3) Line 114: Hodzic et al. (2016) mention that 20% of total NMVOC emissions (not including SVOC emissions) are assumed to be IVOC emissions. Could the authors elaborate more on how SVOC emissions are excluded from total NMVOC emissions and how double counting is avoided when using S/IVOC emissions derived from NMVOC and POA?

S/IVOCs emissions and SOA formation from these are the subjects of the SOA scheme in CAM-chem. There are uncertainties and limitations with CAM-chem OA simulation methods, and improving these are beyond the scope of this study, which aims to develop a new SOA framework in CAM to achieve consistent results between CAM and CAM-chem. Additionally, future improvements in CAM-chem can be easily incorporated into CAM using the method in this study.

However, we understand the reviewer's concern about this double-counting issue in CAM-chem, as it will affect CAM simulations too, since we try to make CAM SOA similar to CAM-chem SOA. Therefore we have decided to add a detailed discussion about S/IVOCs and POA to the manuscript, as follows.

Since there are many uncertainties in OA simulation in models, continuous updates to the CAM-chem VBS scheme will be necessary. As Hodzic et al. (2020) pointed out, CAM-chem showed good agreement in reproducing absolute OA concentrations during the Atmospheric Tomography (ATom) aircraft campaign, but the POA/SOA ratio was overestimated. CAM-chem considers SOA from S/IVOCs based on the assumption that the emission inventory they used reported POA emissions after evaporation to S/IVOCs (Hodzic et al., 2016). However, there is a possibility of double-counting depending on the timing of measuring POA emission flux. Additionally, the assumption that SVOC emissions were included in POA emissions was not sufficiently constrained due to limited observation data (Wu et al., 2019). Fang et al. (2021) reported that IVOCs did not show significant correlations with POA or NMVOCs for on-road vehicles. CAM-chem also assumes a single value for the organic mass to organic carbon (OM/OC) ratio of 1.4 for POA. In contrast, GEOS-Chem has used an OM/OC ratio of 2.1 for POA (Henze et al., 2008; Jo et al., 2013; Hodzic et al., 2020), which would lead to 50% higher POA concentrations than CAM-chem if other conditions are the same. However, observed OM/OC values are spatially and seasonally dependent, typically ranging from 1.3 to 2.5 (Aiken et al., 2008; Philip et al., 2014). These uncertain factors suggest that current assumptions about S/IVOCs and POA may need to be updated in the future. Still, such updates in CAM-chem can be easily transferred into CAM through the consistent framework established in this study.

3.4) Line 160: Could the authors clarify if the SOA yields from CAM-chem used to derive the values reported in the text for the new scheme are derived from Hodzic et al. (2016) Table 2?

Yes, SOA yields in CAM-chem were originally from Hodzic et al. (2016) Table 1 values. We have clarified this in the manuscript as follows.

Second, VBS product yields (forming semi-volatile compounds in the model, sum of gas and aerosol phases, and used for the interactive emissions) have been calculated based on the CAM-chem yields, which were adapted from Hodzic et al. (2016).

3.5) Only low NOx yields are used in the design of the new scheme. Some discussion about how the scheme could be extended to cope with high NOx regimes would be interesting, at least as a perspective for future works.

We have added the sentence in the discussion as follows. However, we believe low  $NO_x$  yields will be sufficient for most studies using CAM, as CAM is primarily used for climate studies rather than focusing on urban air pollution where high  $NO_x$  yields become important.

For studies focusing on urban air quality and resulting climate effects, SOA yields can be changed to high- $NO_x$  yields instead of low- $NO_x$  yields without code changes.

3.6) Apart from the SOA yields, the authors should provide the resulting scaling factors applied to SOA precursor emissions. Figure 1 or Table 1 could be improved by detailing the key parameters of the scheme (e.g. emission scaling factors, H\*, photolysis rate, ...).

SOA yields are scaling factors applied to emissions, as we lump all semivolatiles into a single intermediate species (SOAE). For other key parameters (H\*, photolysis rate, wet scavenging efficiency, etc.) we have added more information in Table 1 and the manuscript (see response to 3.2 for details).

3.7) Line 184: In Section 2.4, the authors could provide some additional details about the linkages between BC, POA and SOA. The aerosols are assumed to be internally mixed in MAM4, is the total mass of one mode used in the gas-aerosol partitioning of SOA or only the total OA mass? How is SOA affecting the microphysical ageing of the mode? The reader would appreciate a brief description instead of simply providing a reference.

Here we have added a detailed description of the linkages between BC, POA, and SOA, instead of giving a simple reference as follows.

Unlike SOA, there is no difference in BC and POA simulation schemes between CAM and CAM-chem, because BC and POA are chemically inert and the standard aerosol module is the same (MAM4) for both CAM and CAM-chem. However, BC and POA can change through the following processes. Both POA and BC are emitted into the primary carbon mode, where they are coated by sulfate and SOA, and then transferred into the accumulation mode and slowly aged through condensation and coagulation, with a threshold coating thickness of eight hygroscopic monolayers of SOA (Liu et al., 2016). In the accumulation mode, aerosols are hydrophilic, with a volume-weighted hygroscopicity calculated based on the volume mixing rule. A strong increase in SOA formation over source regions, which is true for CAM-chem SOA based on Hodzic et al. (2016) SOA scheme, increases the internally mixed aerosol number, which causes enhanced aging of BC and POA. As a result, the CAM SOA scheme simulates more than two times higher primary carbon mode concentrations of BC and POA through reduced aging, but ~10% lower accumulation mode concentrations of both. This results in increased dry deposition and decreased wet deposition in the CAM SOA scheme compared to the CAM-chem SOA scheme, as the primary carbon mode is hydrophobic but the accumulation mode is hydrophilic in CESM. More details can be found in Tilmes et al. (2019).

3.8) Line 223: Additional budget metrics would be appreciated in Table 2 (i.e., production, deposition, lifetime).

We have added deposition, photolytic loss, and lifetime in Table S1, with several sensitivity tests and similarity statistics as follows.

**Table S1.** Global annual budgets and statistics for simulated SOA. Results are based on one year nudged simulation in 2013. The table also includes sensitivity simulation results that demonstrate the effect of excluding a specific aspect of the SOA scheme developed in this study. Three statistics are calculated against CAM-chem results: Normalized Mean Bias (NMB), the fraction of grid cells within a factor of 2 (FO2) and 5 (FO5). Statistics are calculated based on monthly mean grid cell points.

| Simulation case                                                     | Burden
(Gg) | All Loss
(Tg yr -1 ) | Dry dep.
(Tg yr -1 ) | Wet dep.
(Tg yr -1 ) | Photo.
loss
(Tg yr -1 ) | Lifetime
(days) | NMB
(%) | FO2
(%) | FO5
(%) |
|---------------------------------------------------------------------|----------------|------------------------------------|------------------------------------|------------------------------------|------------------------------------------|--------------------|------------|------------|------------|
| CAM-chem                                                            | 1022           | 132                                | 10.1                               | 66.0                               | 55.9                                     | 2.83               | -          | -          | -          |
| CAM6                                                                | 948            | 80                                 | 11.9                               | 68.2                               | 0.0                                      | 4.32               | -7.2       | 24         | 44         |
| CAM
(this study)                                                 | 1027           | 133                                | 7.8                                | 67.5                               | 57.3                                     | 2.83               | 0.5        | 62         | 82         |
| This study
(with CAM6 SOAG
emissions)                         | 318            | 37                                 | 2.4                                | 15.7                               | 18.8                                     | 3.14               | -68.9      | 32         | 70         |
| This study
(without photolytic loss)                             | 2997           | 116                                | 10.5                               | 105.3                              | 0.0                                      | 9.45               | 193.2      | 14         | 27         |
| This study
(with CAM6 saturation vapor
pressure and enthalpy) | 1057           | 138                                | 7.9                                | 70.8                               | 58.8                                     | 2.81               | 3.4        | 60         | 82         |
| This study
(without deposition of SOAG)                          | 1367           | 194                                | 11.3                               | 109.0                              | 74.2                                     | 2.57               | 33.7       | 61         | 86         |
| This study
(with the assumption of 10%
of POA as oxygenated)  | 1126           | 149                                | 9.2                                | 77.6                               | 61.9                                     | 2.77               | 10.2       | 63         | 83         |
| This study
(without intermediate tracer
SOAE)                 | 714            | 117                                | 13.6                               | 69.8                               | 33.9                                     | 2.22               | -30.1      | 16         | 39         |

Technical comments:

3.9) Figure 1: soa3 is repeated in CAM-chem scheme

We have corrected this typo in the figure.

3.10) Line 117: CLM is used in different places in the manuscript, consider defining the acronym like "the Community Land Model (CLM) version 5" here. The same applies for other acronyms like CAM or MEGAN.

We have changed the text as follows.

Biogenic VOCs are calculated online using the Model of Emissions of Gases and Aerosols from Nature (MEGAN) version 2.1 (Guenther et al., 2012) available in the Community Land Model (CLM) version 5

The Community Earth System Model Version 2 (CESM2) has two different SOA schemes, one simplified scheme for the Community Atmosphere Model (CAM) version 6 (Danabasoglu et al., 2020) and the Whole Atmosphere Community Climate Model (WACCM) version 6 with the Middle Atmosphere (MA) chemistry (Gettelman et al., 2019b)

3.11) Line 225: missing capitalizing letter "Gas-phase SOA (SOAG) is substantially..."

Corrected.

**Reviewer #4**

This manuscript gives a generally clear description of the development of a simplified SOA scheme which is designed to mimic as far as possible the results of the more complex scheme (CAM-Chem). I think that the manuscript is worthy of publication after addressing the comments raised by referees. At this stage three referees have already delivered comments, so I will try to avoid making the same points. However, I do have some additional remarks:

4.1) The manuscript would benefit from version numbers or better labels for the original and new schemes. Terms like "NEW" (or "current simplified SOA") tend to age poorly, and somebody reading this manuscript in 10 years time might find that NEW was now OLD, and current wasn't current.

We have changed "OLD" to "CAM6" and "NEW" to CAM (this study)" throughout the entire manuscript, tables (Tables 1, 2, and S1), and figures (Figures 1-5, and S2-S8). As a result, "NEW" is used in the manuscript in only a few places, where we would like to emphasize that this scheme is newly developed. The term "OLD" and "current" have been removed from all occurrences. The text changes are as follows.

[revised manuscript text omitted]

4.2) I miss discussion of these schemes compared to the real world, and indeed the text often makes it sound as though matching the more complex scheme is the same as improving model performance. For example, statements such as "SOA underprediction during the ... winter time", or "simulates too much SOA globally" sounds as though we are comparing with reality, but the comparison is only with the complex scheme. Be explicit with such comparisons, and please add some comparisons with the real world!

The comparisons to observations are outside the scope of this study, as the purpose of developing this new SOA scheme in CAM is to achieve consistent results between different model configurations. Furthermore, even if CAM (both old and new) SOA schemes show better evaluation results compared to the CAM-chem SOA scheme, we believe it would likely stem from incorrect reasoning, as the CAM-chem SOA includes a more detailed scientific basis. The SOA scheme in CAM-chem has been evaluated against observations in previous studies; therefore, we have added a sentence to note the previous evaluations for the SOA scheme used in this study.

The VBS approach in CAM-chem has been evaluated against surface and aircraft observations in the United States, Europe, East Asia, the Amazon, and remote atmosphere (Hodzic et al., 2016, 2020; Tilmes et al., 2019; Jo et al., 2021; Oak et al., 2022).

We have also changed several sentences to avoid statements giving the impression that CAM is compared to observations or true states.

In terms of reproducing CAM-chem SOA, the lower SOA during the Northern Hemisphere winter time and the SOA build-up in the upper atmosphere (< 100 hPa) are greatly improved in this study.

The differences between CAM-chem and CAM are slightly increased over the Tropics for individual SW and LW fluxes

Overall, the SOA scheme in this study shows slight improvements in other latitudes in addition to the Arctic region when it comes to reproducing CAM-chem results.

The reduced differences can be further confirmed by the global spatial distributions shown in Fig. S7, the CAM simulation in this study shows results closer to CAM-chem in most locations globally.

The large differences between CAM-chem and CAM6 for the SW + LW flux over the Arctic from the nudged meteorology simulations (Fig. 4c) are also found in all historical simulations as shown in Fig. 5, for both 1850s and 2000s simulations.

In terms of global averages (Table 2), the CAM SOA scheme in this study also demonstrates improvements in terms of consistency between CAM and CAM-chem

4.3) p2, L20, and elsewhere. It is surprising that a new SOA scheme would affect BC and POA, so I would add a short explanation of why this happens here and in the main text.

**We have added a brief explanation in the abstract as follows.**

Furthermore, other carbonaceous aerosols (black carbon and primary organic aerosol) in CAM6 become closer to CAM-chem results, due to more similar microphysical aging time scales influenced by SOA coating, which in turn leads to comparable wet deposition fluxes.

**And we have added a detailed description of the microphysics effects between BC, POA, and SOA in the main text as follows.**

Unlike SOA, there is no difference in BC and POA simulation schemes between CAM and CAM-chem, because BC and POA are chemically inert and the standard aerosol module is the same (MAM4) for both CAM and CAM-chem. However, BC and POA can change through the following processes. Both POA and BC are emitted into the primary carbon mode, where they are coated by sulfate and SOA, and then transferred into the accumulation mode and slowly aged through condensation and coagulation, with a threshold coating thickness of eight hygroscopic monolayers of SOA (Liu et al., 2016). In the accumulation mode, aerosols are hydrophilic, with a volume-weighted hygroscopicity calculated based on the volume mixing rule. A strong increase in SOA formation over source regions, which is true for CAM-chem SOA based on Hodzic et al. (2016) SOA scheme, increases the internally mixed aerosol number, which causes enhanced aging of BC and POA. As a result, the CAM SOA scheme simulates more than two times higher primary carbon mode concentrations of BC and POA through reduced aging, but ~10% lower accumulation mode concentrations of both. This results in increased dry deposition and decreased wet deposition in the CAM SOA scheme compared to the CAM-chem SOA scheme, as the primary carbon mode is hydrophobic but the accumulation mode is hydrophilic in CESM. More details can be found in Tilmes et al. (2019).

Significant improvements are also found for BC and POA. CAM6 simulates up to ~45% differences while CAM in this study shows up to ~7% differences for BC and POA (Table 2). This is attributed to microphysical aging between different aerosol modes and associated wet deposition processes described in Sect 2.4. As discussed in Tilmes et al. (2019), the CAM6 SOA scheme simulates a higher primary carbon mode (41 and 276 Gg for BC and POA) compared to both CAM-chem (19 and 93 Gg) and the CAM SOA scheme in this study (14 and 81 Gg). Conversely, the CAM6 SOA scheme simulates a lower accumulation mode (90 and 429 Gg for BC and POA) compared to CAM-chem (97 and 494 Gg) and the CAM SOA scheme in this study (97 and 493 Gg).

4.4) p4, L43. The VBS is not the most detailed version of SOA mechanisms; indeed it is rather simple compared to e.g. Xavier e al (2019)

The sentence has been revised to limit the explanation to the case of the 3D atmospheric chemistry models.

The SOA parameterization in 3D atmospheric chemistry models varies from the simple method of multiplying constant yields to emissions, to the rather complex volatility basis set (VBS) approach (Donahue et al., 2006, 2011, 2012; Jimenez et al., 2009)

4.5) p4 L41. I would add that parameterizations are used because there are also many uncertainties in our basic knowledge of SOA formation. Also, why do you say "generally use"? All models use parameterizations!

"generally" is used here because some types of SOA can be simulated without parameterization. For example, IEPOX-SOA can be calculated from the reactive uptake of IEPOX, and IEPOX can be explicitly simulated from the gas-phase reactions of isoprene to isoprene peroxy radical to ISOPOOH to IEPOX. We have added the limited knowledge of SOA formation as another reason for using parameterization in the manuscript.

Atmospheric models generally use parameterizations to simulate SOA because it is composed of a wide range of different organic molecules (Goldstein and Galbally, 2007) and due to limited knowledge of SOA formation in the atmosphere (Nault et al., 2021).

4.6) p7, 2nd paragraph. It isn't clear here if "SOA" means both gas and particle phase, or just particle phase. You describe the gas-phase SOAG as as an "intermediate precursor of SOA" (L111), which makes it sound as though SOAG is lost to particulate SOA (SOAP?) with no return. In the VBS the SOAG and SOAP phase should be in equilibrium, so SOAG is no more an intermediate than SOAP. Throughout the manuscript it is not clear what the term "SOA" means. Please clarify and tighten up the notation.

We agree with the reviewer and modified the text to make sure that we mean the aerosol phase when we refer to SOA, as follows.

dry and wet deposition of gas-phase semivolatiles (SOAGs). Note that dry and wet deposition are applied to SOA in all simulation cases as shown in Fig 1.

the oxidation of those VOCs with OH, O3 and NO3 makes gas phase semivolatiles (SOAG) that

are in equilibrium with SOA according to the volatility bins.

Deposition of gas phase semivolatiles (SOAG) and the photolytic reaction of SOA are also added

SOAG (semivolatiles that are in equilibrium with particle phase SOA),

SOAG is substantially underestimated in both CAM cases

4.7) p7, L114. Do these SVOC and IVOC assumptions add mass to the emissions, or are they just fractions of the original emissions? If 20% of NMVOC are treated as S/IVOC, are they effectively removed from the gas-phase (and ozone producing) mechanisms?

These assumptions add mass to the emissions, as we assume current emission inventories are missing those because they are sticky and difficult to detect. All VBS reactions including S/IVOCs, do not affect gas-phase chemistry. We have clarified this as follows:

VOCs and oxidants are not consumed to avoid duplication, as VOC chemistry is separately simulated in CAM-chem (Jo et al., 2021).

4.8) p8, 145. The new scheme isn't really explained here. Please refer to Fig. 1 (which is quite clear, except whether SOA means SOAP), and that the new scheme uses a 1-bin VBS.

We refer to Fig. 1 here, and we have also added more explanations in the Fig. 1 caption for clarity.

The new SOA scheme uses a similar approach to the current SOA scheme in CAM, but several modifications have been made to allow more consistent results with the VBS scheme in CAM-chem (Fig. 1).

**Figure 1.** The notations are based on variable names used in CESM2. Note that "SOAG" begins with 0, while "soa" starts with 1 in CAM-chem (Tilmes et al., 2019; Emmons et al., 2020). In CESM2, gases are written in upper case and aerosols are written in lower case.

4.9) p8, L135. Is dry and wet deposition not considered for particle-phase SVOC? Fig. 1 suggests it is.

The particle phase SOA undergoes dry and wet deposition in all simulation cases, so we haven't added the text here. Now we have clarified this in the text as follows.

Note that dry and wet deposition are applied to SOA in all simulation cases as shown in Fig 1.

4.10) p8, L150. Add "generally" before yield; many factors other than simple carbon number decide the yields.

We have added "generally".

This change can be scientifically justified because SOA yields generally increase with the carbon number (Lim and Ziemann, 2009; Srivastava et al., 2022).

4.11) p9, I found this first paragraph rather confusing. This text discusses VBS yields as though they are fixed quantities, but do you mean the yield of gas+particle phase compounds (SOAG+SOAP), or the yield of particle phase (SOAP) only? The particle yields depend heavily on the ambient absorbing mass, as well as temperature.

We meant gas+particle phase compounds. Now we use "VBS product yields" instead of "SOA yields" to avoid confusion. We have changed and added texts in this paragraph as follows.

Second, VBS product yields (forming semi-volatile compounds in the model, sum of gas and aerosol phases, and used for the interactive emissions) have been calculated based on the CAM-chem yields, which were adapted from Hodzic et al. (2016). The VBS product yields for the first four bins and 20% of the fifth bin are summed up for each compound. Only 20% of the fifth bin yield is used, as it is the most volatile bin and its saturation vapor pressure is 100 times higher than the bin we use in CAM. We selected 20% based on the SOA burden comparison between CAM-chem and CAM, by adjusting this fraction with multiple simulation tests. We consider VBS product yields from OH reactions only in this calculation, because the reaction with OH is dominant for VOCs. Only low NOx yields are used in this study which is consistent with Tilmes et al. (2019), which is appropriate for global climate studies with 1° horizontal resolution of the model grid. For air quality studies with high spatial resolution, CAM-chem with NOx-dependent SOA yields can be used (Schwantes et al., 2022). The resulting yields derived from CAM-chem results are 0.28, 0.64, 0.04, 0.16, 0.45, 0.35, 0.41, and 0.80 for monoterpenes, sesquiterpenes, isoprene, benzene, toluene, xylenes, IVOC, and SVOC, respectively. These yields are constants and do not change during the run, as in CAM-chem. It is worth noting that

those yields can be easily updated in the CAM run-time namelist file if there is a future update to the CAM-chem VBS scheme.

4.12) p9, L174. Why isn't deposition of particle phase SOA considered? This statement contrasts with Fig.1.

Deposition of particle phase SOA is already considered in the current CAM6 SOA scheme as well; therefore, we did not add this. We only described changes to be made to the CAM6 scheme, so it was missing. We have added a sentence to clarify this as follows.

Deposition of gas phase semivolatiles (SOAG) and the photolytic reaction of SOA are also added (deposition of SOA is already considered in CAM6), which can affect SOA concentrations in the remote atmosphere.

4.13) p18, section 4. The conclusions should give a quick summary of what the "new SOA" scheme entails, e.g. 1-bin etc.

We have added the characteristics of the new SOA scheme as follows.

In this study, we developed a new SOA scheme for use in CAM with simple chemistry. This new SOA scheme was designed to close the gap between CAM and CAM-chem in terms of aerosols and radiative effects while maintaining computational efficiency. The new SOA scheme was derived based on the parameters used in the VBS scheme in CAM-chem, without changing the overall architecture of the simple SOA scheme in CAM6. For instance, VOC species for forming SOA were matched to CAM-chem, an intermediate species was introduced to mimic VOC chemistry, missing loss processes were added, and VBS parameters such as enthalpy of vaporization and saturation vapor pressure were updated. As a result, the computational cost remained almost the same with the new SOA scheme (within the range of computing environment variability).

4.14) p19, L348. "improved the performance of .." = another example of confusing text for me, see point (2).

We have modified this sentence as follows.

while the SOA scheme in this study demonstrated similar variabilities compared to CAM-chem SOA.

4.15) p20. Code and data availability. As already noted by the editor, this manuscript should be associated with a doi where readers can access the code. And which model results are available at DASH. And what is DASH, and where is it? This was very vague!

We have added a Zenodo DOI for the code, and we will use Zenodo for uploading model results instead of DASH (an NCAR repository, which is now called GDEX) once the manuscript is accepted.

4.16) p21-24, References. Many lack doi information.

Now all individual references have DOI information.